# ARE SPIKING NEURAL NETWORKS MORE EXPRESSIVE THAN ARTIFICIAL NEURAL NETWORKS?

## ABSTRACT

This article studies the expressive power of spiking neural networks with firing-time-based information encoding, highlighting their potential for future energy-efficient AI applications when deployed on neuromorphic hardware. The computational power of a network of spiking neurons has already been studied via their capability of approximating any continuous function. By using the Spike Response Model as a mathematical model of a spiking neuron and assuming a linear response function, we delve deeper into this analysis and prove that spiking neural networks generate continuous piecewise linear mappings. We also show that they can emulate any multi-layer (ReLU) neural network with similar complexity. Furthermore, we show that the maximum number of linear regions generated by a spiking neuron scales exponentially with respect to the input dimension, a characteristic that distinguishes it significantly from an artificial (ReLU) neuron. Our results further extend the understanding of the approximation properties of spiking neural networks and open up new avenues where spiking neural networks can be deployed instead of artificial neural networks without any performance loss.

## 1 INTRODUCTION

Despite the remarkable success of deep neural networks (LeCun et al., 2015), the downside of training and inferring on large deep neural networks implemented on classical digital hardware lies in their substantial time and energy consumption (Thompson et al., 2021). The rapid advancement in the field of neuromorphic computing offers both analog and digital computation, energy-efficient computational operations, and faster inference (Schuman et al., 2022), (Christensen et al., 2022). Neuromorphic computers, using (artificial) neurons and synapses aim to replicate the human brain structure and functions (Maan et al., 2017). They are typically programmed with networks of spiking neurons where programs are defined by the structure and parameters of the network rather than explicit instructions (Schuman et al., 2022). These spiking neurons more realistically model neural activity compared to other neuron models, enabling brain-like behavior in these computers.

In traditional artificial neural networks (ANNs), both inputs and outputs are analog-valued. In spiking neural networks (SNNs), neurons transmit information in the form of an *action-potential* or a *spike* (Gerstner et al., 2014). Spikes can be considered as point-like events in time, where incoming spikes received via a neuron's synapses trigger new spikes in the outgoing synapses. This asynchronous information transmission in SNNs differs from ANNs, where information is propagated synchronously through the network. Hence, a key difference between ANNs and SNNs lies in the significance of timing in the operation of SNNs. Moreover, the analog input information needs to be encoded in the form of spikes, necessitating a spike-based encoding scheme.

Different encoding schemes enable spiking neurons to represent analog-valued inputs, broadly categorized into rate and temporal coding (Gerstner & van Hemmen, 1993). Rate coding refers to the number of spikes in a given time period whereas, in temporal coding, the precise timing of spikes matters (Maass, 2001). The notion of firing rate adheres to neurobiological experiments where it was observed that some neurons fire frequently in response to some external stimuli (Stein, 1967), (Gerstner et al., 2014). The latest experimental results indicate that the firing time of a neuron is essential in order for the system to respond faster to more complex sensory stimuli (Hopfield, 1995), (Thorpe et al., 1996), (Abeles, 1991). The firing rate results in higher latency and is computationally expensive due to extra overhead related to temporal averaging. With the firing time, each spike car-

ries a significant amount of information, thus the resulting signal can be quite sparse. While there is no accepted standard on the correct description of neural coding, in this work, we assume that the information is encoded in the firing time of a neuron. The event-driven, sparse information propagation, as seen in time-to-first-spike encoding (Gerstner & Kistler, 2002), facilitates system efficiency in terms of reduced computational power and improved energy efficiency.

It is clear that the differences in the processing of information between ANNs and SNNs should also lead to differences in the computations performed by these models. Several groups have analyzed the expressive power of ANNs (Yarotsky, 2017), (Cybenko, 1989), (Gühring et al., 2020), (Petersen & Voigtlaender, 2018), and in particular provided explanations for the superior performance of deep networks over shallow ones (Daubechies et al., 2022), (Yarotsky, 2017). In the case of ANNs with ReLU activation function, the number of linear regions into which the input space is partitioned into is another property that highlights the advantages of deep networks over shallow ones. Unlike shallow networks, deep networks divide the input space into exponentially more number of linear regions (Goujon et al., 2022), (Montúfar et al., 2014) enabling them to express more complex functions. There exists further approaches to characterize the expressiveness of ANNs, e.g., the concept of VC-dimension in the context of classification problems (Bartlett et al., 1999), (Goldberg & Jerrum, 1995), (Bartlett et al., 2019). In this work, we consider the problem of expressiveness from the perspective of approximation theory and by quantifying the number of linear regions.

Few attempts have been made that aim to understand the computational power of SNNs. In Maass (1996b), Maass (1996c), it has been shown that spiking neurons can emulate Turing machines, arbitrary threshold circuits, and sigmoidal neurons in temporal coding. In Maass (1996a), biologically relevant functions are depicted that can be emulated by a single spiking neuron but require complex ANNs to achieve the same task. A common theme is that the model of spiking neurons and the description of their dynamics varies, i.e., they are chosen and adjusted with respect to a specific goal or task. Moraitis et al. (2021), Jeffares et al. (2022) provide intriguing insights on superior expressive power of SNNs over ANNs, showcasing their ability to excel in certain settings of temporal input. Moraitis et al. (2018), Izhikevich (2006) show that spiking neurons can represent multiple variables simultaneously leading to performance gains in contrast to ANNs. Stöckl & Maass (2021) aims at generating high-performance SNNs for image classification using a modified spiking neuron model that limits the number of spikes emitted by each neuron while considering precise spike timing. The primary challenge in advancing the domain of SNNs has revolved around devising training methodologies. The typical approach is to either train from scratch (Lee et al., 2020), (Wu et al., 2018), (Comsa et al., 2020), (Göltz et al., 2021) or convert trained ANNs into SNNs performing the same tasks (Rueckauer et al., 2017), (Kim et al., 2018), (Rueckauer & Liu, 2021), (Stanojevic et al., 2022a), (Stanojevic et al., 2022b), (Rueckauer & Liu, 2018). The latter works concentrate on the algorithmic construction of SNNs approximating or emulating given ANNs. In an attempt to follow up along the lines of previous works, in particular, the expressivity of SNNs (Maass, 1996a), the linear region property of ANNs (Montúfar et al., 2014) as well as first strides in that direction in SNNs (Mostafa, 2018), and the conversion of ReLU-ANNs into SNNs with time-to-first-spike encoding (Stanojevic et al., 2022b), we aim to extend the theoretical understanding that characterizes the differences and similarities in the expressive power between a network of spiking and artificial neurons employing a piecewise-linear activation function. Our aim is to assess if SNNs under the Spike Response Model match the expressiveness of ANNs in approximating different function spaces under given complexity bounds and in terms of the number of linear regions they can generate.

**Contributions** In this paper, to analyze SNNs, we employ the noise-free version of the Spike Response Model (SRM) (Gerstner, 1995). It describes the state of a neuron as a weighted sum of response and threshold functions. We assume a linear response function, where additionally each neuron spikes at most once to encode information through precise spike timing. The spiking networks based on linear SRM are succinctly referred to as LSRM (see Remark 1). This in turn simplifies the model and also makes the mathematical analysis more feasible for larger networks as compared to other neuronal models where the spike dynamics are described in the form of differential equations. In the future, we aim to expand our investigation to encompass multi-spike responses and refractoriness effects, thus, the selection of this model is appropriate and comprehensive. The main results are centered around the comparison of expressive power between LSRMs and ANNs:

- **Equivalence of Approximation:** We prove that LSRMs outputs a continuous piecewise linear mapping. Moreover, we construct a two-layer LSRM that emulates the ReLU non-

linearity. Then, we extend the construction to multi-layer neural networks and show that an LSRMs has the capacity to effectively reproduce the output of any (ReLU) ANN. Furthermore, we present explicit complexity bounds that are essential for constructing an LSRMs capable of realizing an equivalent ANN. We also provide insights on the influence of the encoding scheme and the impact of different parameters on the above approximation results. These findings imply that LSRMs can approximate any function as accurately as deep ANNs with piecewise linear activation function.

- **Linear Regions:** We demonstrate that the maximum number of linear regions that a one-layer LSRM generates scales exponentially with input dimension. This suggests that a shallow LSRM can be as expressive as a deep ReLU network in terms of the number of linear regions required to express certain types of continuous piecewise linear functions. This is a characteristic of LSRM neurons that sets it apart from a ReLU neuron, thereby illustrating differences in the structure of computations between LSRMs and ANNs.

**Impact**    The theoretical findings presented herein deepen our understanding of the differences and similarities between the expressive power of ANNs and SNNs. In theory, our findings prove that the potential low-power neuromorphic implementation of LSRMs is an energy-efficient alternative to the computation performed by (ReLU-)ANNs without loss of expressive power. Moreover, it also enhances our understanding of performing computations where time plays a critical role. We anticipate that the advances in event-driven neuromorphic computing will have a tremendous impact, especially for edge-computing applications such as robotics, autonomous driving etc. This is accomplished while prioritizing energy efficiency — a crucial factor in modern computing landscapes.

**Outline**    In Section 2, we introduce necessary definitions, including spiking neural networks under the Spike Response Model. We present our main results in Section 3. In Section 4, we discuss related work and conclude in Section 5 by summarizing the limitations and implications of our results. The proofs of all the results are provided in the Appendix A.

## 2    SPIKING NEURAL NETWORKS

In neuroscience literature, several mathematical models exist that describe the generation and propagation of action-potentials. Action-potentials or spikes are short electrical pulses that are the result of electrical and biochemical properties of a biological neuron (Gerstner et al., 2014). We refer to Gerstner et al. (2014) for a comprehensive and detailed introduction to the dynamics of spiking neurons. To study the expressivity of SNNs, the main principles of a spiking neuron are condensed into a (simplified) mathematical model, where certain details about the biophysics of a biological neuron are neglected. Following Maass (1996b), we consider the Spike Response Model (SRM) (Gerstner, 1995) as a formal model for a spiking neuron. It effectively captures the dynamics of the Hodgkin-Huxley model (Kistler et al., 1997), (Gerstner et al., 2014), the most accurate model in describing neuronal dynamics, and is a generalized version of the leaky integrate and fire model (Gerstner, 1995). The SRM leads to the subsequent definition of an SNN (Maass, 1996c).

**Definition 1.** *A* spiking neural network $\Phi$ *under the SRM is a (simple) finite directed graph* $(V, E)$ *and consists of a finite set $V$ of spiking neurons, a subset $V_{in} \subset V$ of input neurons, a subset $V_{out} \subset V$ of output neurons, and a set $E \subset V \times V$ of synapses. Each synapse $(u, v) \in E$ is associated with a synaptic weight $w_{uv} \geq 0$, a synaptic delay $d_{uv} \geq 0$, and a response function $\varepsilon_{uv} : \mathbb{R}^+ \to \mathbb{R}$. Each neuron $v \in V \setminus V_{in}$ is associated with a firing threshold $\theta_v > 0$, and a membrane potential $P_v : \mathbb{R} \to \mathbb{R}$, which is given by*

$$P_v(t) = \sum_{(u,v) \in E} \sum_{t_u^f \in F_u} w_{uv} \varepsilon_{uv}(t - t_u^f), \tag{1}$$

*where $F_u = \{t_u^f : 1 \leq f \leq n \text{ for some } n \in \mathbb{N}\}$ denotes the set of firing times of a neuron $u$, i.e., times $t$ whenever $P_u(t)$ reaches $\theta_u$ from below.*

In general, the membrane potential also includes the *threshold function* $\Theta_v : \mathbb{R}^+ \to \mathbb{R}^+$, that models the refractoriness effect. That is, if a neuron $v$ emits a spike at time $t_v^f$, $v$ cannot fire again for some time interval immediately after $t_v^f$, regardless of how large its potential might be. However, we assume that each neuron fires at most once, i.e., information is encoded in the firing time of single

spikes. Thus, in Definition 1, the refractoriness effect can be ignored and the contribution of $\Theta_v$ is modelled by the constant $\theta_v$. Moreover, the single spike condition simplifies (1) to

$$P_v(t) = \sum_{(u,v)\in E} w_{uv}\varepsilon_{uv}(t - t_u), \quad \text{where } t_u = \inf\left\{t \geq \min_{(z,u)\in E}\{t_z + d_{zu}\} : P_u(t) \geq \theta_u\right\}. \quad (2)$$

The *response function* $\varepsilon_{uv}$ models the impact of a spike from a presynaptic neuron $u$ on the membrane potential of a postsynaptic neuron $v$ (Gerstner, 1995). A biologically realistic approximation of $\varepsilon_{uv}$ is a delayed $\alpha$ function (Gerstner, 1995), which is non-linear and leads to intractable problems when analyzing the propagation of spikes through an SNN. Hence, following Maass (1996b), we consider a simplified response and only require $\varepsilon_{uv}$ to satisfy the following condition:

$$\varepsilon_{uv}(t) = \begin{cases} 0, & \text{if } t \notin [d_{uv}, d_{uv} + \delta], \\ s \cdot (t - d_{uv}), & \text{if } t \in [d_{uv}, d_{uv} + \delta], \end{cases} \quad \text{where } s \in \{+1, -1\} \text{ and } \delta > 0. \quad (3)$$

The parameter $\delta$ is some constant assumed to be the length of a linear segment of the response function. The variable $s$ reflects the fact that biological synapses are either *excitatory* or *inhibitory* and the *synaptic delay* $d_{uv}$ is the time required for a spike to travel from $u$ to $v$. Inserting condition (3) in (2) and setting $w_{uv} := s \cdot w_{uv}$, i.e., allowing $w_{uv}$ to take arbitrary values in $\mathbb{R}$, yields

$$P_v(t) = \sum_{(u,v)\in E} \mathbf{1}_{\{0 < t - t_u - d_{uv} \leq \delta\}} w_{uv}(t - t_u - d_{uv}). \quad (4)$$

**Remark 1.** *Note that we denote spiking neural networks (SNNs) based on the simplified SRM model discussed above as LSRMs and the corresponding spiking neurons as LSRM neurons.*

## 2.1 COMPUTATION IN TERMS OF FIRING TIME

Using (4) enables us to iteratively compute the firing time $t_v$ of each neuron $v \in V \setminus V_{\text{in}}$ if we know the firing time $t_u$ of each neuron $u \in V$ with $(u,v) \in E$ by solving for $t$ in

$$\inf_{t \geq \min_{(u,v)\in E}\{t_u + d_{uv}\}} P_v(t) = \inf_{t \geq \min_{(u,v)\in E}\{t_u + d_{uv}\}} \sum_{(u,v)\in E} \mathbf{1}_{\{0 < t - t_u - d_{uv} \leq \delta\}} w_{uv}(t - t_u - d_{uv}) = \theta_v. \quad (5)$$

Set $E(\mathbf{t}_U) := \{(u,v) \in E : d_{uv} + t_u < t_v \leq d_{uv} + t_u + \delta\}$, where $\mathbf{t}_U := (t_u)_{(u,v)\in E}$ is a vector containing the given firing times of the presynaptic neurons. The firing time $t_v$ satisfies

$$\theta_v = \sum_{(u,v)\in E} \mathbf{1}_{\{0 < t_v - t_u - d_{uv} \leq \delta\}} w_{uv}(t_v - t_u - d_{uv}) = \sum_{(u,v)\in E(\mathbf{t}_U)} w_{uv}(t_v - t_u - d_{uv}), \quad (6)$$

$$\text{i.e., } t_v = \frac{\theta_v}{\sum_{(u,v)\in E(\mathbf{t}_U)} w_{uv}} + \frac{\sum_{(u,v)\in E(\mathbf{t}_U)} w_{uv}(t_u + d_{uv})}{\sum_{(u,v)\in E(\mathbf{t}_U)} w_{uv}}. \quad (7)$$

Here, $E(\mathbf{t}_U)$ identifies the presynaptic neurons that actually have an effect on $t_v$ based on $\mathbf{t}_U$. For instance, if $t_w > t_v$ for some synapse $(w,v) \in E$, then $w$ did not contribute to the firing of $v$ since the spike from $w$ arrived after $v$ already fired so that $(w,v) \notin E(\mathbf{t}_U)$. Equation (7) shows that $t_v$ is a weighted sum (up to a positive constant) of the firing times of neurons $u$ with $(u,v) \in E(\mathbf{t}_U)$. Flexibility, i.e., non-linearity, in this model is provided through the variation of the set $E(\mathbf{t}_U)$. Depending on the firing time of the presynaptic neurons $\mathbf{t}_U$ and the associated parameters (weights, delays, threshold), $E(\mathbf{t}_U)$ contains a set of different synapses so that $t_v$ via (7) alters accordingly. The dynamics of a neuron in this model is depicted in Figure 1.

Subsequently, we will employ (7) to analyze and construct LSRMs. In particular, we simply assume that the length $\delta$ of the linear segment of the response function introduced in (3) is large enough so that (7) holds. Informally, a small linear segment requires incoming spikes to have a correspondingly small time delay to jointly affect the potential of a neuron. Otherwise, the impact of the earlier spikes on the potential may already have vanished before the subsequent spikes arrive. Consequently, incorporating $\delta$ as an additional parameter in the LSRM model leads to additional complexity since the same firing patterns may result in different outcomes. However, an in-depth analysis of this effect is left as future work. When the parameter $\delta$ is large, the simplified linear Spike Response Model, stemming from the linear response function and the constraint of single-spike dynamics, exhibits similarities to the integrate and fire model. Conversely, if $\delta$ is small, it resembles the leaky integrate and fire model. The final model obtained provides a highly simplified version of the dynamics observed in biological neural systems. Nevertheless, we attain a theoretical model that, in principle, can be directly implemented on neuromorphic hardware and moreover, enables us to analyze the computations that can be carried out by a network of spiking neurons.

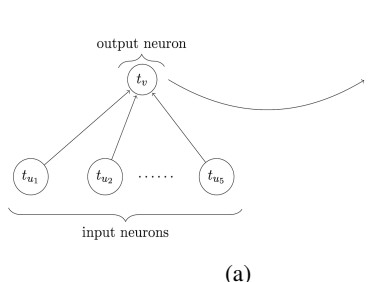
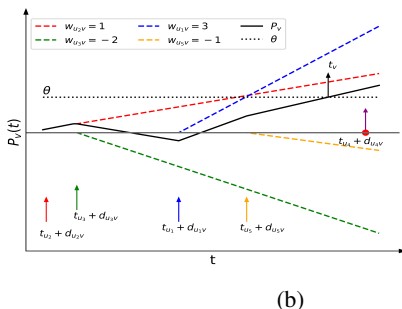

(a)

(b)

Figure 1: (a) A spiking neuron $v$ with five input neurons $u_1, \ldots, u_5$ that fire at times $t_{u_1}, \ldots, t_{u_5}$, respectively. (b) The trajectory in black shows the evolution of the membrane potential $P_v(t)$ of $v$ as a result of incoming spikes (vertical arrows). Neurons $u_1$ and $u_2$ generate positive responses, whereas neurons $u_3$ and $u_5$ trigger negative response, with the response magnitudes denoted by $w_{u_i v}$. The spike from neuron $u_4$ does not influence the firing time $t_v$ of $v$ since $t_v < t_{u_4} + d_{u_4 v}$.

## 2.2 INPUT AND OUTPUT ENCODING

By restricting our framework of LSRMs to acyclic graphs, we can arrange the underlying graph in layers and equivalently represent LSRMs by a sequence of their parameters. This is analogous to the common representation of feedforward ANNs via a sequence of matrix-vector tuples (Berner et al., 2022), (Petersen & Voigtlaender, 2018).

**Definition 2.** *Let $L \in \mathbb{N}$. An (LSRM) $\Phi$ associated to the acyclic graph $(V, E)$ is a sequence of matrix-matrix-vector tuples*

$$\Phi = ((W^1, D^1, \Theta^1), (W^2, D^2, \Theta^2), \ldots, (W^L, D^L, \Theta^L))$$

*where $N_0, \ldots, N_L \in \mathbb{N}$ and each $W^l \in \mathbb{R}^{N_{l-1} \times N_l}$, $D^l \in \mathbb{R}_+^{N_{l-1} \times N_l}$, and $\Theta^l \in \mathbb{R}_+^{N_l}$. The matrix entries $W_{uv}^l$ and $D_{uv}^l$ represent the weight and delay value associated with the synapse $(u, v) \in E$, respectively, and the entry $\Theta_v^l$ is the firing threshold associated with node $v \in V$. $N_0$ is the input dimension and $N_L$ is the output dimension of $\Phi$. We call $N(\Phi) := \sum_{j=0}^L N_j$ the number of neurons and $L(\Phi) := L$ denotes the number of layers of $\Phi$.*

**Remark 2.** *In an ANN, the input signal is propagated in a synchronized manner layer-wise through the network (see Definition 5). In contrast, in any SNN, information is transmitted via spikes, where spikes from layer $l - 1$ affect the membrane potential of layer $l$ neurons, resulting in asynchronous propagation due to variable firing times among neurons.*

To employ SNNs, the (typically analog) input information needs to be encoded in the firing times of the neurons in the input layer, and similarly, the firing times of the output neurons need to be translated back to an appropriate target domain. We will refer to this process as input encoding and output decoding. The applied encoding scheme certainly depends on the specific task at hand and the potential power and suitability of different encoding schemes is a topic that warrants separate investigation on its own. Our focus in this work lies on exploring the intrinsic capabilities of LSRMs, rather than the specifics of the encoding scheme. Thus, we can formulate some guiding principles for establishing a reasonable encoding scheme. First, the firing times of input and output neurons should encode analog information in a consistent way so that different networks can be concatenated in a well-defined manner. This enables us to construct suitable subnetworks and combine them appropriately to solve more complex tasks. Second, in the extreme case, the encoding scheme might directly contain the solution to a problem, underscoring the need for a sufficiently simple and broadly applicable encoding scheme to avoid this.

**Definition 3.** *Let $[a, b]^d \subset \mathbb{R}^d$ and $\Phi$ be a LSRM with input neurons $u_1, \ldots, u_d$ and output neurons $v_1, \ldots, v_n$. Fix reference times $T_{in} \in \mathbb{R}^d$ and $T_{out} \in \mathbb{R}^n$. For any $x \in [a, b]^d$, we set the firing times of the input neurons to $(t_{u_1}, \ldots, t_{u_d})^T = T_{in} + x$ and the corresponding firing times of the output neurons $(t_{v_1}, \ldots, t_{v_n})^T = T_{out} + y$, determined via (7), encode the target $y \in \mathbb{R}^n$.*

**Remark 3.** *A bounded input range ensures that appropriate reference times can be fixed. Note that the introduced encoding scheme translates analog information into input firing times in a continuous*

*manner. Occasionally, we will point out the effect of adjusting the scheme and we will sometimes with a slight abuse of notation refer to $T_{in}, T_{out}$ as one-dimensional objects, i.e., $T_{in}, T_{out} \in \mathbb{R}$ which is justified if the corresponding vectors contain the same element in each dimension.*

For the discussion ahead, we distinguish between a network and the target function it realizes. A network is a structured set of weights, delays and thresholds as defined in Definition 2, and the target function it realizes is the result of the asynchronous propagation of spikes through the network.

**Definition 4.** *On $[a, b]^d \subset \mathbb{R}^d$, the realization of an LSRM $\Phi$ with output neurons $v_1, \ldots, v_n$ and reference times $T_{in} \in \mathbb{R}^d$ and $T_{out} \in \mathbb{R}^n$, where $T_{out} > T_{in}$, is defined as the map $\mathcal{R}_\Phi : \mathbb{R}^d \to \mathbb{R}^n$,*

$$\mathcal{R}_\Phi(x) = -T_{out} + (t_{v_1}, \ldots, t_{v_n})^T.$$

Next, we give a corresponding definition of an ANN and its realization.

**Definition 5.** *Let $L \in \mathbb{N}$. An artificial neural network $\Psi$ is a sequence of matrix-vector tuples*

$$\Psi = ((W^1, B^1), (W^2, B^2), \ldots, (W^L, B^L)),$$

*where $N_0, \ldots, N_L \in \mathbb{N}$ and each $W^l \in \mathbb{R}^{N_{l-1} \times N_l}$ and $B^l \in \mathbb{R}^{N_l}$. $N_0$ and $N_L$ are the input and output dimension of $\Psi$. We call $N(\Psi) := \sum_{j=0}^{L} N_j$ the number of neurons of the network $\Psi$, $L(\Psi) := L$ the number of layers of $\Psi$ and $N_l$ the width of $\Psi$ in layer $l$. The realization of $\Psi$ with component-wise activation function $\sigma : \mathbb{R} \to \mathbb{R}$ is defined as the map $\mathcal{R}_\Psi : \mathbb{R}^{N_0} \to \mathbb{R}^{N_L}$, $\mathcal{R}_\Psi(x) = y_L$, where $y_L$ results from*

$$y_0 = x, \quad y_l = \sigma(W^l y_{l-1} + B^l), \ \text{for } l = 1, \ldots, L-1, \quad \text{and } y_L = W^L y_{L-1} + B^L. \quad (8)$$

In the remainder, we always employ the ReLU activation function $\sigma(x) = \max(0, x)$. One can perform basic actions on neural networks such as concatenation and parallelization to construct larger networks from existing ones. Adapting a general approach for ANNs as defined in Berner et al. (2022), Petersen & Voigtlaender (2018), we formally introduce the concatenation and parallelization of networks of spiking neurons in the Appendix A.1.

## 3 MAIN RESULTS

First, we prove that LSRMs generate **C**ontinuous **P**iece**w**ise **L**inear (CPWL) mappings, followed by realizing a ReLU activation function using a two-layer LSRM. Subsequently, we show that LSRMs can emulate the realization of any multi-layer ANN employing ReLU as an activation function. Lastly, we analyze the number of linear regions generated by LSRMs and compare the arising pattern to the well-studied case of ReLU-ANNs. If not stated otherwise, the encoding scheme introduced in Definition 3 is applied and the results need to be understood with respect to this specific encoding.

### 3.1 LSRMS REALIZE CONTINUOUS PIECEWISE LINEAR MAPPING

A broad class of ANNs based on a wide range of activation functions such as ReLU generate CPWL mappings (Dym et al., 2020), (DeVore et al., 2021). In other words, these ANNs partition the input domain into regions, the so-called linear regions, on which an affine function represents the neural network's realization. We show that LSRMs also express CPWL mappings under very general conditions. The proof of the statement can be found in the Appendix A.2.

**Theorem 1.** *Any LSRM $\Phi$ realizes a CPWL function provided that the sum of synaptic weights of each neuron is positive and the encoding scheme is a CPWL function.*

**Remark 4.** *Note that the encoding scheme introduced in Definition 3 is a CPWL mapping. The positivity of the sum of weights ensures that each neuron in the network emits a spike, in particular it is a sufficient but not necessary condition to guarantee that spikes are emitted by the output neurons. In general, if the positivity condition is not met by a neuron, then it does not fire for certain inputs. Therefore, the case may arise where an output neuron does not fire and the realization of the network is not well-defined. One could adapt the definition of the realization of an LSRM, however, the CPWL property described in the theorem may be lost.*

## 3.2 EQUIVALENCE OF APPROXIMATION

Despite the fact that ReLU is a very basic CPWL function, it is not straightforward to realize ReLU via LSRMs; see Appendix A.3 for the proof.

**Theorem 2.** *Let $a < 0 < b$. There does not exist a one-layer LSRM that realizes $\sigma(x) = \max(0, x)$ on $[a, b]$. However, $\sigma$ can be realized by a two-layer LSRM on $[a, b]$.*

**Remark 5.** *We note that the encoding scheme that converts the analog values into the time domain plays a crucial role. In the proof of the Theorem 2, an LSRM is constructed that realizes $\sigma$ via the encoding scheme $T_{in} + \cdot$ and $T_{out} + \cdot$. At the same time, the encoding scheme $T_{in} - \cdot$ and $T_{out} - \cdot$ fails in the two-layer case, whereas utilizing an inconsistent input and output encoding enables us to construct a one-layer LSRM that realizes $\sigma$. This shows that not only the network but also the applied encoding scheme is highly relevant. For details, we refer to Appendix A.3. Moreover, in a hypothetical real-world implementation, which certainly includes some noise, the constructed LSRM that realizes ReLU is not necessarily robust with respect to input perturbation. Analyzing the behaviour and providing error estimations is an important future task.*

Next, we extend the realization of a ReLU neuron to the entire network, i.e., realize the output of any ReLU network using LSRMs. Please refer to Appendix A.4 for detailed proof.

**Theorem 3.** *Let $L, d \in \mathbb{N}$, $[a, b]^d \subset \mathbb{R}^d$ and let $\Psi$ be an arbitrary ANN of depth $L$ and fixed width $d$ employing a ReLU non-linearity, and having a one-dimensional output. Then, there exists an LSRM $\Phi$ with $N(\Phi) = N(\Psi) + L(2d + 3) - (2d + 2)$ and $L(\Phi) = 3L - 2$ that realizes $\mathcal{R}_\Psi$ on $[a, b]^d$.*

**Remark 6.** *The result can be generalized to ANNs with varying widths that employ any type of piecewise linear activation function. Additionally, the complexity of an LSRM can be captured in other ways than in terms of the number of computational units and layers, e.g., the total number of spikes emitted in LSRMs is related to its energy consumption since emitting spikes consumes energy. Hence, the minimum number of spikes to realize a given function class may be a reasonable complexity measure with regard to energy efficiency for SNNs. Further research in this direction is necessary to evaluate the complexity of LSRMs via different measures with their benefits and drawbacks.*

It is well known that ReLU-ANNs not only realize CPWL mappings but that every CPWL function can be represented by ReLU-ANNs if no restrictions are placed on the number of parameters or the depth of the networks (Arora et al., 2018), (Daubechies et al., 2022). Thus, ReLU-ANNs can represent any LSRM with a CPWL encoding scheme. In contrast, our results also imply that LSRMs can represent every ReLU-ANN and thereby every CPWL function. The key difference in the realization of arbitrary CPWL mappings is the necessary size and complexity of the respective ANN and LSRM. Recall that realizing ReLU via LSRMs required more computational units than the corresponding ANN (see Theorem 3). Conversely, we demonstrate using a toy example that LSRMs can realize certain CPWL functions with fewer number of computational units and layers than ReLU-ANNs.

**Example 1.** *For $a < 0 < \theta < b$, consider the CPWL function $f : [a, b] \rightarrow \mathbb{R}$ given by*

$$f(x) = -\frac{1}{2}\sigma(-x - \theta) - \frac{1}{2}\sigma(-x + \theta) = -\frac{1}{2}\max(-x - \theta, 0) - \frac{1}{2}\max(-x + \theta, 0). \quad (9)$$

*A one-layer LSRM with one output unit and two input units can realize $f$. However, any ReLU-ANN requires at least two layers and four computational units to realize $f$; see Appendix A.5 for the proof.*

These observations illustrate that the computational structure of LSRMs differs significantly from that of ReLU-ANNs, while neither model is clearly beneficial in terms of network complexity to express all CPWL functions. To gain a better understanding of this divergent behaviour, in the next section, we study the number of linear regions that LSRMs generate.

## 3.3 NUMBER OF LINEAR REGIONS

The number of linear regions can be seen as a measure for the flexibility and expressivity of the corresponding CPWL function. Similarly, we can measure the expressivity of an ANN by the number of linear regions of its realization. The connection of the depth, width, and activation function of an ANN to the maximum number of its linear regions is well-established, e.g., with increasing depth the

number of linear regions can grow exponentially in the number of parameters of an ANN (Montúfar et al., 2014), (Arora et al., 2018), (Goujon et al., 2022). This property offers one possible explanation for why deep networks tend to outperform shallow networks in expressing complex functions. Can we observe a similar behaviour for LSRMs? To that end, we first analyze the properties of a spiking neuron. For the proof, we refer to Appendix A.2.

**Theorem 4.** *Let $\Phi$ be a one-layer LSRM with a single output neuron $v$ and $d$ input neurons $u_1, \ldots, u_d$ such that $\sum_{i=1}^{d} w_{u_i v} > 0$. Then $\Phi$ partitions the input domain into at most $2^d - 1$ linear regions. In particular, for a sufficiently large input domain, the maximal number of linear regions is attained if and only if all synaptic weights are positive.*

**Remark 7.** *The parameters of $\Phi$ determine the number of linear regions into which the input domain is divided by $\Phi$. In particular, if $w_{u_j v} \leq 0$, then $u_j$ can not cause the firing of $v$ and $\Phi$ can not achieve the maximal number of linear regions. Similarly, one can derive via (7) that any subset of input neurons $\{u_{j_1}, \ldots, u_{j_k}\}$ with net negative weights did not cause a firing of $v$. The linear region corresponding to a subset of input neurons with a positive sum of weights is actually realized if the input domain is suitably large. Finally, the condition $\sum_{i=1}^{d} w_{u_i v} > 0$ ensures that the notion of linear region is well-defined. Otherwise, the input domain is still partitioned into polytopes by $\Phi$ but there exists a region where the realization of the network is not well-defined (see Remark 4).*

A one-layer ReLU-ANN with one output neuron will partition the input domain into at most two linear regions, independent of the dimension of the input. In contrast, for a one-layer LSRM with one output neuron, the maximum number of linear regions scales exponentially in the input dimension. In the case of LSRMs, non-linearity is the intrinsic property of the model and emerges from the subset of neurons that have an effect on the firing time of the output neuron, whereas in the ANN it is applied on the single output neuron. By shifting the non-linearity and applying it to the input, the ANN could exhibit the same exponential scaling of the linear regions as the LSRM. However, this change has rather a detrimental effect on the expressivity since the partitioning of the input domain is fixed and independent of the parameters of the ANN. The flexibility of LSRMs to generate arbitrary linear regions is to a certain extent limited, albeit not entirely restricted as in the adjusted ANN. For LSRMs one can explicitly compute the boundaries of the linear regions. This is exemplarily demonstrated for a two-dimensional input space in Appendix A.2. It turns out that only specific hyperplanes are eligible as boundaries of the linear region in this simple scenario. The full power of ANN comes into play with large numbers of layers, however, our result in Theorem 4 suggests that a shallow LSRM can be as expressive as a deep ReLU network in terms of the number of linear regions required to express certain types of CPWL functions. In Dym et al. (2020), the authors showed that a deep ANN employing any piecewise linear activation function cannot span all CPWL functions with the number of linear regions scaling exponentially in the number of parameters of the network. Studying these types of functions and identifying (or excluding) similar behaviour for LSRMs requires a deeper analysis of the capabilities of LSRMs, providing valuable insights into their computational power. This aspect is left for future investigation.

## 4 RELATED WORK

In this section, we mention the most relevant results that investigate the computational or expressive power of SNNs. One of the central results in this direction is the Universal Approximation Theorem for LSRMs (Maass, 1995), demonstrating the existence of LSRMs that approximates arbitrary feedforward ANNs employing sigmoidal activation function and thus, approximating any continuous function. In contrast, we show that LSRMs can realize arbitrary ANNs with CPWL activation and further specify the size of the network to achieve the associated realization, which has not been previously demonstrated. Moreover, we also study the expressivity of LSRMs in terms of the number of linear regions and provide new insights on realizations generated by LSRMs. Comsa et al. (2020) showed that continuous functions can be approximated to arbitrary precision using SRM in temporal coding. In Zhang & Zhou (2022), the authors investigate self-connection SNNs, demonstrating their capacity to efficiently approximate discrete dynamical systems. Our approach centers on precise spike timing, while theirs hinges on firing rates and includes a distinct model featuring self-connections, further setting their approach apart from ours. A connection between SNNs and PWL functions was already noted in Mostafa (2018). The author showed that a spiking network consisting of non-leaky integrate and fire neurons, employing exponentially decaying synaptic current kernels and temporal coding, exhibits a PWL input-output relation after a transformation of the

time variable. This piecewise relation is continuous unless small perturbations influences the spiking behaviour, specifically concerning whether the neuron fires or remains inactive.

Another line of research focuses on converting trained ANNs into equivalent SNNs and, thereby avoiding or facilitating the training process of SNNs. This has been studied for various spike patterns, encoding schemes and spiking neuron models (Stöckl & Maass, 2021), (Stanojevic et al., 2022b), (Kim et al., 2018), (Rueckauer et al., 2017), (Rueckauer & Liu, 2021), (Yousefzadeh et al., 2019), (Rueckauer & Liu, 2018), (Zhang et al., 2019), (Yan et al., 2021). By introducing an algorithmic conversion from ANNs to SNNs, one also establishes approximation or emulation capabilities of SNNs in the considered setting. Most related to our analysis are the results in Stanojevic et al. (2022b). Under certain assumptions, the authors define a one-to-one neuron mapping that converts a trained ReLU network to a corresponding SNN consisting of integrate and fire neurons by a nonlinear transformation of parameters. However, significant distinctions exist between the approaches, particularly in terms of the chosen model, objectives, and methodology. Our choice of the model is driven by our intention to better understand expressivity outcomes. In terms of methodology, we introduce an auxiliary neuron to ensure the firing of neurons even when a corresponding ReLU neuron exhibits zero activity. This diverges from their approach, which employs external current and a special parameter to achieve similar outcomes. Moreover, our work involves a fixed threshold for neuron firing, whereas their model incorporates a threshold that varies with time. Lastly, we study the differences in the structure of computations between ANNs and SNNs, whereas in Stanojevic et al. (2022b), only the conversion of ANNs to SNNs is examined and not vice versa.

## 5 DISCUSSION

The central aim of this paper is to study and compare the expressive power of SNNs and ANNs employing any piecewise linear activation function. In an ANN, information is propagated across the network in a synchronized manner. In contrast, in SNNs, spikes are only emitted once a subset of neurons in the previous layer triggers a spike in a neuron in the subsequent layer. Hence, the imperative role of time in biological neural systems accounts for differences in computation between SNNs and ANNs. Our expressivity result in Theorem 3 implies that LSRMs can essentially approximate any function with the same accuracy and (asymptotic) complexity bounds as (deep) ANNs employing a piecewise linear activation function, given the response function satisfies some basic assumptions. Rather than approximating some function space by emulating a known construction for ReLU networks, one could also achieve optimal approximations by leveraging the intrinsic capabilities of LSRMs instead. The findings in Theorem 4 indicate that the latter approach may indeed be beneficial in terms of the complexity of the architecture in certain circumstances. However, we point out that finding optimal architectures for approximating different classes of functions is not the focal point of our work. The significance of our results lies in investigating theoretically the approximation and expressivity capabilities of SNNs, highlighting their potential as an alternative computational model for complex tasks. Extending the model of a LSRM neuron by incorporating, e.g., multiple spikes of a neuron, may yield an improvement on our results. However, by increasing the complexity of the model the analysis also tends to be more elaborate. In the aforementioned case of multiple spikes the threshold function becomes important so that additional complexity when approximating some target function is introduced since one would have to consider the coupled effect of response and threshold functions. Similarly, the choice of the response function and the frequency of neuron firings will surely influence the approximation results and we leave this for future work.

**Limitations** We prove that LSRMs are as expressive as ReLU-ANNs in theory. However, achieving similar results in practice heavily relies on the effectiveness of the employed training algorithms. The implementation of efficient learning algorithms with weights, delays and thresholds as programmable parameters is left for future work. In this work, our choice of model resides on theoretical considerations and not on practical considerations regarding implementation. However, there might be other models of spiking neurons that are more apt for implementation purposes — see e.g. Stanojevic et al. (2022b) and Comsa et al. (2020). Furthermore, in reality, due to the ubiquitous sources of noise in the spiking neurons, the firing activity of a neuron is not deterministic. For mathematical simplicity, we perform our analysis in a noise-free case. Generalizing to the case of noisy spiking neurons is important (for instance with respect to the aforementioned implementation in noisy environments) and may lead to further insights in the model.

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

# A    APPENDIX

**Outline**   We start by introducing the spiking network calculus in Section A.1 to compose and parallelize different networks. In Section A.2, we show that LSRMs output CPWL functions and establish a relation between the input dimension of an LSRM and the number of linear regions. The proof of Theorem 2 is given in Section A.3. Finally, in Section A.4, we prove that an LSRM can realize the output of any ReLU network.

## A.1    SPIKING NEURAL NETWORK CALCULUS

It can be observed from Definition 3 that both inputs and outputs of LSRMs are encoded in a unified format. This characteristic is crucial for concatenating/parallelizing two spiking network architectures that further enable us to attain compositions/parallelizations of network realizations.

We operate in the following setting: Let $L_1$, $L_2$, $d_1$, $d_2$, $d_1'$, $d_2' \in \mathbb{N}$. Consider two LSRMs $\Phi_1$, $\Phi_2$ given by

$$\Phi_i = ((W_1^i, D_1^i, \Theta_1^i), \dots, (W_{L_i}^i, D_{L_i}^i, \Theta_{L_i}^i)), \quad i = 1, 2,$$

with input domains $[a_1, b_1]^{d_1} \subset \mathbb{R}^{d_1}$, $[a_2, b_2]^{d_2} \subset \mathbb{R}^{d_2}$ and output dimension $d_1', d_2'$, respectively. Denote the input neurons by $u_1, \dots, u_{d_i}$ with respective firing times $t_{u_j}^i$ and the output neurons by $v_1, \dots, v_{d_i'}$ with respective firing times $t_{v_j}^i$ for $i = 1, 2$. By Definition 3, we can express the firing times of the input neurons as

$$t_u^1(x) := (t_{u_1}^1, \dots, t_{u_{d_1}}^1)^T = T_{\text{in}}^1 + x \quad \text{for } x \in [a_1, b_1]^{d_1},$$
$$t_u^2(x) := (t_{u_1}^2, \dots, t_{u_{d_2}}^2)^T = T_{\text{in}}^2 + x \quad \text{for } x \in [a_2, b_2]^{d_2} \tag{10}$$

and, by Definition 4, the realization of the networks as

$$\mathcal{R}_{\Phi_1}(x) = -T_{\text{out}}^1 + t_v^1(t_u^1(x)) := -T_{\text{out}}^1 + (t_{v_1}^1, \dots, t_{v_{d_1'}}^1)^T \quad \text{for } x \in [a_1, b_1]^{d_1},$$
$$\mathcal{R}_{\Phi_2}(x) = -T_{\text{out}}^2 + t_v^2(t_u^2(x)) := -T_{\text{out}}^2 + (t_{v_1}^2, \dots, t_{v_{d_2'}}^2)^T \quad \text{for } x \in [a_2, b_2]^{d_2} \tag{11}$$

for some constants $T_{\text{in}}^1 \in \mathbb{R}^{d_1}$, $T_{\text{in}}^2 \in \mathbb{R}^{d_2}$, $T_{\text{out}}^1 \in \mathbb{R}^{d_1'}$, $T_{\text{out}}^2 \in \mathbb{R}^{d_2'}$.

We define the concatenation of the two networks in the following way.

**Definition 6.** *(Concatenation) Let $\Phi_1$ and $\Phi_2$ be such that the input layer of $\Phi_1$ has the same dimension as the output layer of $\Phi_2$, i.e., $d_2' = d_1$. Then, the concatenation of $\Phi_1$ and $\Phi_2$, denoted as $\Phi_1 \bullet \Phi_2$, represents the $(L_1 + L_2)$-layer network*

$$\Phi_1 \bullet \Phi_2 := ((W_1^2, D_1^2, \Theta_1^2), \dots, (W_{L_2}^2, D_{L_2}^2, \Theta_{L_2}^2), (W_1^1, D_1^1, \Theta_1^1), \dots, (W_{L_1}^1, D_{L_1}^1, \Theta_{L_1}^1)).$$

**Lemma 1.** *Let $d_2' = d_1$ and fix $T_{in} = T_{in}^2$ and $T_{out} = T_{out}^1$. If $T_{out}^2 = T_{in}^1$ and $\mathcal{R}_{\Phi_2}([a_2, b_2]^{d_2}) \subset [a_1, b_1]^{d_1}$, then*

$$\mathcal{R}_{\Phi_1 \bullet \Phi_2}(x) = \mathcal{R}_{\Phi_1}(\mathcal{R}_{\Phi_2}(x)) \quad \text{for all } x \in [a, b]^{d_2}$$

*with respect to the reference times $T_{in}, T_{out}$. Moreover, $\Phi_1 \bullet \Phi_2$ is composed of $N(\Phi_1) + N(\Phi_2) - d_1$ computational units.*

*Proof.* It is straightforward to verify via the construction that the network $\Phi_1 \bullet \Phi_2$ is composed of $N(\Phi_1) + N(\Phi_2) - d_1$ computational units. Moreover, under the given assumptions $\mathcal{R}_{\Phi_1} \circ \mathcal{R}_{\Phi_2}$ is well-defined so that (10) and (11) imply

$$
\begin{aligned}
\mathcal{R}_{\Phi_1 \bullet \Phi_2}(x) &= -T_{\text{out}} + t_v^1(t_v^2(T_{\text{in}} + x)) = -T_{\text{out}}^1 + t_v^1(t_v^2(T_{\text{in}}^2 + x)) = -T_{\text{out}}^1 + t_v^1(t_v^2(t_u^2(x))) \\
&= -T_{\text{out}}^1 + t_v^1(T_{\text{out}}^2 + \mathcal{R}_{\Phi_2}(x)) = -T_{\text{out}}^1 + t_v^1(T_{\text{in}}^1 + \mathcal{R}_{\Phi_2}(x)) \\
&= -T_{\text{out}}^1 + t_v^1(t_u^1(\mathcal{R}_{\Phi_2}(x))) = \mathcal{R}_{\Phi_1}(\mathcal{R}_{\Phi_2}(x)) \quad \text{for } x \in [a_2, b_2]^{d_2}.
\end{aligned}
$$

$\square$

In addition to concatenating networks, we also perform parallelization operation on LSRMs.

**Definition 7.** *(Parallelization) Let $\Phi_1$ and $\Phi_2$ be such that they have the same depth and input dimension, i.e., $L_1 = L_2 =: L$ and $d_1 = d_2 =: d$. Then, the parallelization of $\Phi_1$ and $\Phi_2$, denoted as $P(\Phi_1, \Phi_2)$, represents the $L$-layer network with $d$-dimensional input*

$$
P(\Phi_1, \Phi_2) := ((\tilde{W}_1, \tilde{D}_1, \tilde{\Theta}_1), \dots, (\tilde{W}_L, \tilde{D}_L, \tilde{\Theta}_L)),
$$

*where*

$$
\tilde{W}_1 = \begin{pmatrix} W_1^1 & W_1^2 \end{pmatrix}, \quad \tilde{D}_1 = \begin{pmatrix} D_1^1 & D_1^2 \end{pmatrix}, \quad \tilde{\Theta}_1 = \begin{pmatrix} \Theta_1^1 \\ \Theta_1^2 \end{pmatrix}
$$

*and*

$$
\tilde{W}_l = \begin{pmatrix} W_l^1 & 0 \\ 0 & W_l^2 \end{pmatrix}, \quad \tilde{D}_l = \begin{pmatrix} D_l^1 & 0 \\ 0 & D_l^2 \end{pmatrix}, \quad \tilde{\Theta}_l = \begin{pmatrix} \Theta_l^1 \\ \Theta_l^2 \end{pmatrix}, \quad \text{for } 1 < l \le L.
$$

**Lemma 2.** *Let $d := d_2 = d_1$ and fix $T_{in} := T_{in}^1$, $T_{out} := (T_{out}^1, T_{out}^2)$, $a := a_1$ and $b := b_1$. If $T_{in}^2 = T_{in}^1$, $T_{out}^2 = T_{out}^1$ and $a_1 = a_2$, $b_1 = b_2$, then*

$$
\mathcal{R}_{P(\Phi_1, \Phi_2)}(x) = (\mathcal{R}_{\Phi_1}(x), \mathcal{R}_{\Phi_2}(x)) \quad \text{for } x \in [a, b]^d
$$

*with respect to the reference times $T_{in}, T_{out}$. Moreover, $P(\Phi_1, \Phi_2)$ is composed of $N(\Phi_1) + N(\Phi_2) - d$ computational units.*

*Proof.* The number of computational units is an immediate consequence of the construction. Since the input domains of $\Phi_1$ and $\Phi_2$ agree, (10) and (11) show that

$$
\begin{aligned}
\mathcal{R}_{P(\Phi_1, \Phi_2)}(x) &= -T_{\text{out}} + (t_v^1(T_{\text{in}} + x), t_v^2(T_{\text{in}} + x)) = (-T_{\text{out}}^1 + t_v^1(T_{\text{in}}^1 + x), -T_{\text{out}}^2 + t_v^2(T_{\text{in}}^2 + x)) \\
&= (-T_{\text{out}}^1 + t_v^1(t_u^1(x)), -T_{\text{out}}^2 + t_v^2(t_u^2(x))) = (\mathcal{R}_{\Phi_1}(x), \mathcal{R}_{\Phi_2}(x)) \quad \text{for } x \in [a, b]^d.
\end{aligned}
$$

$\square$

**Remark 8.** *Note that parallelization and concatenation can be straightforwardly extended (recursively) to a finite number of networks. Additionally, more general forms of parallelization and concatenations of networks, e.g., parallelization of networks with different depths, can be established. However, for the constructions presented in this work, the introduced notions suffice.*

## A.2 REALIZATIONS OF LSRMS

In this section, we analyze the realization of LSRMs. We show that a LSRM neuron with arbitrarily many input neurons generates a CPWL mapping and establish a correspondence between the input dimension of the LSRM neurons and the number of linear regions of the associated realization. For simplicity, we perform the analysis without employing an encoding scheme of analog values in the time domain via the firing time of the input neurons. However, it is straightforward to incorporate the encoding into the analysis. Moreover, since we show that the firing time of a LSRM neuron is a CPWL function on the input domain, it immediately follows that any LSRM neuron with a CPWL encoding scheme, e.g., as defined in Definition 3, realizes a CPWL mapping. The final step is to extend the analysis from a single LSRM neuron to a network of LSRM neurons.

### A.2.1 Spiking neuron with two inputs

First, we provide a simple toy example to demonstrate the dynamics of a LSRM neuron. Let $v$ be a LSRM neuron with two input neurons $u_1, u_2$. Denote the associated weights and delays by $w_{u_i v} \in \mathbb{R}$ and $d_{u_i v} \geq 0$, respectively, and the threshold of $v$ by $\theta_v > 0$. A spike emitted from $v$ could then be caused by either $u_1$ or $u_2$ or a combination of both. Each possibility corresponds to a linear region in the input space $\mathbb{R}^2$. We consider each case separately under the assumption that $\delta$ in (3) is arbitrarily large and we discuss the implications of this assumption in more detail after presenting the different cases.

**Case 1**: $u_1$ causes $v$ to spike before a potential effect from $u_2$ reaches $v$. Note that this can only happen if $w_{u_1 v} > 0$ and

$$t_{u_2} + d_{u_2 v} \geq t_v = \frac{\theta_v}{w_{u_1 v}} + t_{u_1} + d_{u_1 v},$$

where we applied (6) and (7), and $t_z$ represents the firing time of a neuron $z$. Solving for $t_{u_2}$ leads to

$$t_{u_2} \geq \frac{\theta_v}{w_{u_1 v}} + t_{u_1} + d_{u_1 v} - d_{u_2 v}.$$

**Case 2**: An analogous calculation shows that

$$t_{u_2} \leq -\frac{\theta_v}{w_{u_2 v}} + t_{u_1} + d_{u_1 v} - d_{u_2 v},$$

whenever $u_2$ causes $v$ to spike before a potential effect from $u_1$ reaches $v$.

**Case 3**: The remaining possibility is that spikes from $u_1$ and $u_2$ influence the firing time of $v$. Then, the following needs to hold: $w_{u_1 v} + w_{u_2 v} > 0$ and

$$t_{u_1} + d_{u_1 v} < t_v = \frac{\theta_v}{w_{u_1 v} + w_{u_2 v}} + \sum_i \frac{w_{u_i v}}{w_{u_1 v} + w_{u_2 v}}(t_{u_i} + d_{u_i v}) \quad \text{and}$$

$$t_{u_2} + d_{u_2 v} < t_v = \frac{\theta_v}{w_{u_1 v} + w_{u_2 v}} + \sum_i \frac{w_{u_i v}}{w_{u_1 v} + w_{u_2 v}}(t_{u_i} + d_{u_i v}).$$

This yields

$$t_{u_2} \begin{cases} > -\frac{\theta_v}{w_{u_2 v}} + t_{u_1} + d_{u_1 v} - d_{u_2 v}, & \text{if } \frac{w_{u_2 v}}{w_{u_1 v} + w_{u_2 v}} > 0 \\ < -\frac{\theta_v}{w_{u_2 v}} + t_{u_1} + d_{u_1 v} - d_{u_2 v}, & \text{if } \frac{w_{u_2 v}}{w_{u_1 v} + w_{u_2 v}} < 0 \end{cases},$$

respectively

$$t_{u_2} \begin{cases} < \frac{\theta_v}{w_{u_1 v}} + t_{u_1} + d_{u_1 v} - d_{u_2 v}, & \text{if } \frac{w_{u_1 v}}{w_{u_1 v} + w_{u_2 v}} > 0 \\ > \frac{\theta_v}{w_{u_1 v}} + t_{u_1} + d_{u_1 v} - d_{u_2 v}, & \text{if } \frac{w_{u_1 v}}{w_{u_1 v} + w_{u_2 v}} < 0 \end{cases}.$$

**Example 2.** *In a simple setting with $\theta_v = w_{u_i v} = d_{u_2 v} = 1$ and $d_{u_1 v} = 2$, the above considerations imply the following firing time of $v$ on the corresponding linear regions (see Figure 2):*

$$t_v = \begin{cases} t_{u_1} + 3, & \text{if } t_{u_2} \geq t_{u_1} + 2 \\ t_{u_2} + 2, & \text{if } t_{u_2} \leq t_{u_1} \\ \frac{1}{2}(t_{u_1} + t_{u_2}) + 2, & \text{if } t_{u_1} < t_{u_2} < t_{u_1} + 2 \end{cases}.$$

Already this simple setting with two-dimensional inputs provides crucial insights. The actual number of linear regions in the input domain corresponds to the parameter of the LSRM neuron $v$. In particular, the maximum number of linear regions, i.e. three, can only occur if both weights $w_{u_i v}$ are positive. Similarly, $v$ does not fire at all if both weights are non-positive. The exact number of linear regions depends on the sign and magnitude of the weights. Furthermore, note that the linear regions are described by hyperplanes of the form

$$t_{u_2} \lessgtr t_{u_1} + C_{p,u}, \tag{12}$$

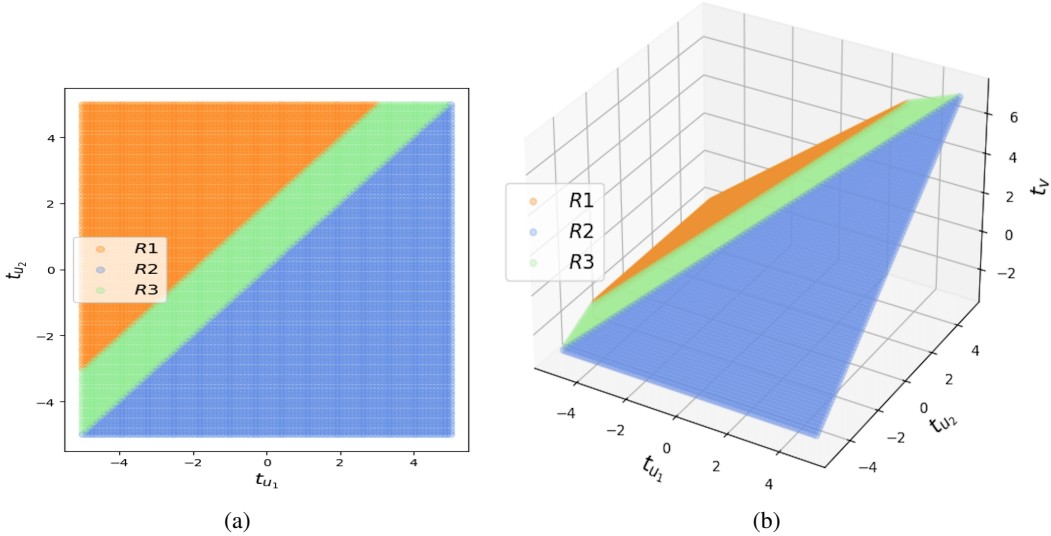

Figure 2: Illustration of Example 2. It shows that the output firing time $t_v(t_{u_1}, t_{u_2})$ as a function of inputs $t_{u_1}, t_{u_2}$ is a CPWL mapping. (a) An illustration of the partitioning of the input space into three different regions. (b) Each region is associated with an affine-linear mapping.

where $C_{p,u}$ is a constant depending on the parameter $p$ corresponding to $v$, i.e., threshold, delays and weights, and the actual input neuron(s) causing $v$ to spike. Hence, $p$ has only a limited effect on the boundary of a linear region; depending on their exact value, the parameter only introduces an additive constant shift.

**Remark 9.** *Dropping the assumption that $\delta$ is arbitrarily large in* (3) *yields an evolved model which is also biologically more realistic. The magnitude of $\delta$ describes the duration in which an incoming spike influences the membrane potential of a neuron. By setting $\delta$ arbitrarily large, we generally consider an incoming spike to have a lasting effect on the membrane potential. Specifying a fixed $\delta$ increases the importance of the timing of the individual spikes as well as the choice of the parameter. For instance, inputs from certain regions in the input domain may not trigger a spike any more since the combined effect of multiple delayed incoming spikes is neglected. An in-depth analysis of the influence of $\delta$ is left as future work and we continue our analysis under the assumption that $\delta$ is arbitrarily large.*

### A.2.2 SPIKING NEURON WITH ARBITRARILY MANY INPUTS

A significant observation in the two-dimensional case is that the firing time $t_v(t_{u_1}, t_{u_2})$ as a function of the input $t_{u_1}, t_{u_2}$ is a CPWL mapping. Indeed, each linear region is associated with an affine linear mapping and crucially these affine mappings agree at the breakpoints. This intuitively makes sense since a breakpoint marks the event when the effect of an additional neuron on the firing time of $v$ needs to be taken into consideration or, equivalently, a neuron does not contribute to the firing of $v$ any more. However, in both circumstances, the effective contribution of this specific neuron is zero (and the contribution of the other neuron remains unchanged) at the breakpoint so that the crossing of a breakpoint and the associated change of a linear region does not result in a discontinuity. We can straightforwardly extend the insights of the two-dimensional to a $d$-dimensional input domain.

Formally, the class of CPWL functions describes functions that are globally continuous and locally linear on each polytope in a given finite decomposition of $\mathbb{R}^d$ into polytopes. We refer to the polytopes as linear regions. First, we assess the number of regions the input domain is partitioned by a spiking neuron.

**Proposition 1.** *Let $v$ be a LSRM neuron with $d$ input neurons $u_1, \ldots, u_d$. Then $\mathbb{R}^d$ is partitioned by $v$ into at most $2^d - 1$ regions.*

*Proof.* The maximum number of regions directly corresponds to $E(\mathbf{t}_U)$ defined in (7). Recall that $E(\mathbf{t}_U)$ identifies the presynaptic neurons that based on their firing times $\mathbf{t}_U = (t_{u_i})_{i=1}^d$ triggered the firing of $v$ at time $t_v$. Therefore, each region in the input domain is associated to a subset of input neurons that is responsible for the firing of $v$ on this specific domain. Hence, the number of regions is bounded by the number of non-empty subsets of $\{u_1, \ldots, u_d\}$, i.e., $2^d - 1$. $\qquad\square$

**Remark 10.** *Observe that any subset of input neurons causes a spike in $v$ if and only if the sum of their weights is positive. Otherwise, the corresponding input region either does not exist or inputs from the corresponding region do not trigger a spike in $v$ since they can not increase the potential $P_v(t)$ as their net contribution is negative, i.e., the potential does not reach the threshold $\theta_v$. Hence, the maximal number of regions is attained if and only if all weights are positive and thereby the sum of weights of any subset of input neurons is positive as well.*

**Remark 11.** *The observations with regard to the parameter $\delta$ in Remark 9 directly transfer from the two- to the $d$-dimensional setting.*

Next, we show that a LSRM neuron generates a CPWL mapping.

**Theorem 5.** *Let $v$ be a LSRM neuron with $d$ input neurons $u_1, \ldots, u_d$. The firing time $t_v(t_{u_1}, \ldots, t_{u_d})$ as a function of the firing times $t_{u_1}, \ldots, t_{u_d}$ is a CPWL mapping provided that $\sum_{i=1}^d w_{u_i v} > 0$, where $w_{u_i v} \in \mathbb{R}$ is the synaptic weight between $u_i$ and $v$.*

*Proof.* The condition $\sum_{i=1} w_{u_i v} > 0$ simply ensures that the input domain is decomposed into regions associated with subsets of input neurons with positive net weight. Hence, the situation described in Remark 10 can not arise and the notion of a CPWL mapping on $\mathbb{R}^d$ is well-defined. Denote the associated delays by $d_{u_i v} \geq 0$ and the threshold of $v$ by $\theta_v > 0$. We distinguish between the $2^d - 1$ variants of input combinations that can cause a firing of $v$. Let $I \subset \{1, \ldots, d\}$ be a non-empty subset and $I^c$ the complement of $I$ in $\{1, \ldots, d\}$, i.e., $I^c = \{1, \ldots, d\} \setminus I$. Assume that all $u_i$ with $i \in I$ contribute to the firing of $v$ whereas spikes from $u_i$ with $i \in I^c$ do not influence the firing of $v$. Then $\sum_{i \in I} w_{u_i v}$ is required to be positive, and by (6) and (7) the following holds:

$$t_{u_k} + d_{u_k v} \geq t_v = \frac{\theta_v}{\sum_{i \in I} w_{u_i v}} + \sum_{i \in I} \frac{w_{u_i v}}{\sum_{j \in I} w_{u_j v}} (t_{u_i} + d_{u_i v}) \quad \text{for all } k \in I^c \qquad (13)$$

and

$$t_{u_k} + d_{u_k v} < t_v = \frac{\theta_v}{\sum_{i \in I} w_{u_i v}} + \sum_{i \in I} \frac{w_{u_i v}}{\sum_{j \in I} w_{u_j v}} (t_{u_i} + d_{u_i v}) \quad \text{for all } k \in I. \qquad (14)$$

Rewriting yields

$$t_{u_k} \geq \frac{\theta_v}{\sum_{i \in I} w_{u_i v}} + \sum_{i \in I} \frac{w_{u_i v}}{\sum_{j \in I} w_{u_j v}} (t_{u_i} + d_{u_i v}) - d_{u_k v} \quad \text{for all } k \in I^c \qquad (15)$$

and

$$t_{u_k} \begin{cases} < \frac{\theta_v}{\sum_{j \in I \setminus k} w_{u_j v}} + \sum_{i \in I \setminus k} \frac{w_{u_i v}}{\sum_{j \in I \setminus k} w_{u_j v}} (t_{u_i} + d_{u_i v}) - d_{u_k v}, & \text{if } \frac{\sum_{i \in I \setminus k} w_{u_i v}}{\sum_{i \in I} w_{u_i v}} > 0 \\ > \frac{\theta_v}{\sum_{j \in I \setminus k} w_{u_j v}} + \sum_{i \in I \setminus k} \frac{w_{u_i v}}{\sum_{j \in I \setminus k} w_{u_j v}} (t_{u_i} + d_{u_i v}) - d_{u_k v}, & \text{if } \frac{\sum_{i \in I \setminus k} w_{u_i v}}{\sum_{i \in I} w_{u_i v}} < 0 \end{cases} \quad \forall k \in I.$$

It is now clear that the firing time $t_v(t_{u_1}, \ldots, t_{u_d})$ as a function of the input $t_{u_1}, \ldots, t_{u_d}$ is a piecewise linear mapping on polytopes decomposing $\mathbb{R}^d$. To show that the mapping is additionally continuous, we need to assess $t_v(t_{u_1}, \ldots, t_{u_d})$ on the breakpoints. Let $I, J \subset \{1, \ldots, d\}$ be index sets corresponding to input neurons $\{u_i : i \in I\}, \{u_j : j \in J\}$ that cause $v$ to fire on the input region $R^I \subset \mathbb{R}^d$, $R^J \subset \mathbb{R}^d$ respectively. Assume that it is possible to transition from $R^I$ to $R^J$ through a breakpoint $t^{I,J} = (t_{u_1}^{I,J}, \ldots, t_{u_d}^{I,J}) \in \mathbb{R}^d$ without leaving $R^I \cup R^J$. Crossing the breakpoint is equivalent to the fact that the input neurons $\{u_i : i \in I \setminus J\}$ do not contribute to the firing of $v$ anymore and the input neurons $\{u_i : i \in J \setminus I\}$ begin to contribute to the firing of $v$.

Assume first that $J \subset I$. Then, we observe that the breakpoint $t^{I,J}$ is necessarily an element of the linear region corresponding to the index set with smaller cardinality, i.e., $t^{I,J} \in R^J$. This is an immediate consequence of (14) and the fact that $t^{I,J}$ is characterized by

$$t_{u_k}^{I,J} + d_{u_k v} = t_v(t^{I,J}) \quad \text{for all } k \in I \setminus J. \qquad (16)$$

Indeed, if $t_{u_k}^{I,J} + d_{u_k v} > t_v(t^{I,J})$, then there exists $\varepsilon_k > 0$ such that (15) also holds for $t_{u_k}^{I,J} \pm \varepsilon$, where $0 \leq \varepsilon < \varepsilon_k$, i.e., a small change in $t_{u_k}^{I,J}$ is not sufficient to change the corresponding linear region, contradicting our assumption that $t^{I,J}$ is a breakpoint.

The firing time $t_v(t^{I,J})$ is explicitly given by

$$t_v(t^{I,J}) = \frac{\theta_v}{\sum_{i \in J} w_{u_i v}} + \sum_{i \in J} \frac{w_{u_i v}}{\sum_{j \in J} w_{u_j v}} (t_{u_i}^{I,J} + d_{u_i v})$$

Using (16), we obtain

$$0 = -\frac{w_{u_k v}}{\sum_{j \in J} w_{u_j v}} (t_v(t^{I,J}) - (t_{u_k}^{I,J} + d_{u_k v})) \quad \text{for all } k \in I \setminus J$$

so that

$$t_v(t^{I,J}) = \frac{\theta_v}{\sum_{i \in J} w_{u_i v}} + \sum_{i \in J} \frac{w_{u_i v}}{\sum_{j \in J} w_{u_j v}} (t_{u_i}^{I,J} + d_{u_i v}) - \sum_{i \in I \setminus J} \frac{w_{u_i v}}{\sum_{j \in J} w_{u_j v}} (t_v(t^{I,J}) - (t_{u_i}^{I,J} + d_{u_i v})).$$

Solving for $t_v(t^{I,J})$ yields

$$t_v(t^{I,J}) = \Big(1 + \sum_{i \in I \setminus J} \frac{w_{u_i v}}{\sum_{j \in J} w_{u_j v}}\Big)^{-1} \cdot \Big(\frac{\theta_v}{\sum_{i \in J} w_{u_i v}} + \sum_{i \in I} \frac{w_{u_i v}}{\sum_{j \in J} w_{u_j v}} (t_{u_i}^{I,J} + d_{u_i v})\Big)$$

$$= \sum_{i \in J} \frac{w_{u_i v}}{\sum_{j \in I} w_{u_j v}} \cdot \Big(\frac{\theta_v}{\sum_{i \in J} w_{u_i v}} + \sum_{i \in I} \frac{w_{u_i v}}{\sum_{j \in J} w_{u_j v}} (t_{u_i}^{I,J} + d_{u_i v})\Big)$$

$$= \frac{\theta_v}{\sum_{i \in I} w_{u_i v}} + \sum_{i \in I} \frac{w_{u_i v}}{\sum_{j \in I} w_{u_j v}} (t_{u_i}^{I,J} + d_{u_i v}),$$

which is exactly the expression for the firing time on $R^I$. This shows that $t_v(t_{u_1}, \ldots, t_{u_d})$ is continuous in $t^{I,J}$. Since the breakpoint $t^{I,J}$ was chosen arbitrarily, $t_v(t_{u_1}, \ldots, t_{u_d})$ is continuous at any breakpoint.

The case $I \subset J$ follows analogously. It remains to check the case when neither $I \subset J$ nor $J \subset I$. To that end, let $i^* \in I \setminus J$ and $j^* \in J \setminus I$. Assume without loss of generality that $t^{I,J} \in R^I$ so that (13) and (14) imply

$$t_{u_{i^*}}^{I,J} + d_{u_{i^*} v} < t_v(t^{I,J}) \leq t_{u_{j^*}}^{I,J} + d_{u_{j^*} v}.$$

Hence, there exists $\varepsilon > 0$ such that

$$t_{u_{i^*}}^{I,J} + d_{u_{i^*} v} < t_{u_{j^*}}^{I,J} + d_{u_{j^*} v} - \varepsilon. \tag{17}$$

Moreover, due to the fact that $t^{I,J}$ is a breakpoint we can find $t^J \in R^J \cap \mathcal{B}(t^{I,J}; \frac{\varepsilon}{3})$, where $\mathcal{B}(t^{I,J}; \frac{\varepsilon}{3})$ denotes the open ball with radius $\frac{\varepsilon}{3}$ centered at $t^{I,J}$. In particular, this entails that

$$-\frac{\varepsilon}{3} < (t_{u_{i^*}}^J - t_{u_{i^*}}^{I,J}), (t_{u_{j^*}}^{I,J} - t_{u_{j^*}}^J) < \frac{\varepsilon}{3},$$

and therefore together with (17)

$$t_{u_{i^*}}^J + d_{u_{i^*} v} - (t_{u_{j^*}}^J + d_{u_{j^*} v}) = (t_{u_{i^*}}^J - t_{u_{i^*}}^{I,J}) + (t_{u_{i^*}}^{I,J} + d_{u_{i^*} v} - (t_{u_{j^*}}^{I,J} + d_{u_{j^*} v})) + (t_{u_{j^*}}^{I,J} - t_{u_{j^*}}^J)$$

$$< 0, \quad \text{i.e., } t_{u_{i^*}}^J + d_{u_{i^*} v} < t_{u_{j^*}}^J + d_{u_{j^*} v}.$$

However, (13) and (14) require that

$$t_{u_{j^*}}^J + d_{u_{j^*} v} < t_v(t^J) \leq t_{u_{i^*}}^J + d_{u_{i^*} v}$$

since $t^J \in R^J$. Thus, $t^{I,J}$ can not exist and the case when neither $I \subset J$ nor $J \subset I$ can not arise. $\qquad\square$

**Remark 12.** *We want to highlight some similarities and differences between two- and $d$-dimensional inputs. In both cases, the actual number of linear regions depends on the choice of parameter, in particular, the synaptic weights. However, the $d$-dimensional case allows for more flexibility in the structure of the linear regions. Recall that in the two-dimensional case, the boundary of any linear region is described by hyperplanes of the form (12). This does not hold if $d > 2$, see e.g. (15). Here, the weights also affect the shape of the linear region. Refining the connection between the boundaries of a linear region, its response function and the specific choice of parameter requires further considerations.*

An interesting question is what effect width and depth has on the realization of an LSRM and, in particular, how the number of linear regions scales with the increasing width and depth of the network. The former problem can be straightforwardly tackled. Any LSRM realizes a CPWL function under very general conditions; see Theorem 1.

***Proof of Theorem 1***. In Theorem 5, we showed that the firing time of a LSRM neuron with arbitrarily many input neurons is a CPWL function with respect to the input under the assumption that the sum of its weight is positive. Since $\Phi$ consists of LSRM neurons arranged in layers it immediately follows that each layer realizes a CPWL mapping. Thus, as a composition of CPWL mappings $\Phi$ itself realizes a CPWL function provided that the input and output encoding are also CPWL functions. □

While Theorem 1 together with Proposition 1 and Remark 10 immediately yield Theorem 4, i.e., the number of linear regions scales at most as $2^d - 1$ in the input dimension $d$ of a LSRM neuron and the number is indeed attained under certain conditions, it is not immediate to obtain a non-trivial upper bound even in the simple case of a one-layer LSRM $\Phi$ with $d_{\text{in}}$ input neurons and $d_{\text{out}}$ output neurons as the following example shows.

**Example 3.** *Via Theorem 4, we certainly can upper bound the number of linear regions generated by $\Phi$ by $(2^{d_{in}} - 1)^{d_{out}}$, i.e., the product of the number of linear regions generated by each individual output neuron. Unfortunately, the bound is far from optimal. Consider the case when $d_{in} = d_{out} = 2$. Then, the structure of the linear regions generated by the individual output neurons is given in (12). In particular, the boundary of the linear regions are described by a set of specific hyperplanes with common normal vector, where the parameter of the LSRM only induce a shift of the hyperplanes. In other words, the hyperplanes separating the linear regions are parallel. Hence, each output neuron generates at most two parallel hyperplanes yielding three linear regions independently (see Figure 2). The number of linear regions generated by the LSRM with two output neurons is therefore given by the number of regions four parallel hyperplanes can decompose the input domain into, i.e., at most $5 < 9 = (2^{d_{in}} - 1)^{d_{out}}$.*

## A.3 Realizing ReLU with LSRMs

**Proposition 2.** *Let $c_1 \in \mathbb{R}$, $c_2 \in (a, b) \subset \mathbb{R}$ and consider $f_1, f_2 : [a, b] \to \mathbb{R}$ defined as*

$$f_1(x) = \begin{cases} x + c_1 & , \text{ if } x > c_2 \\ c_1 & , \text{ if } x \leq c_2 \end{cases} \quad or \quad f_2(x) = \begin{cases} x + c_1 & , \text{ if } x < c_2 \\ c_1 & , \text{ if } x \geq c_2 \end{cases}.$$

*There does not exist a one-layer LSRM with output neuron $v$ and input neuron $u_1$ such that $t_v(x) = f_i(x)$, $i = 1, 2$, on $[a, b]$, where $t_v(x)$ denotes the firing time of $v$ on input $t_{u_1} = x$.*

*Proof.* First, note that a one-layer LSRM realizes a CPWL function. For $c_2 \neq 0$, $f_i$ is not continuous and therefore can not be emulated by the firing time of any one-layer LSRM. Hence, it is left to consider the case $c_2 = 0$. If $u_1$ is the only input neuron, then $v$ fires if and only if $w_{u_1 v} > 0$ and by (7) the firing time is given by

$$t_v(x) = \frac{\theta}{w_{u_1 v}} + x + d_{u_1 v} \quad \text{for all } x \in [a, b],$$

i.e., $t_v \neq f_i$. Therefore, we introduce auxiliary input neurons $u_2, \ldots, u_n$ and assume without loss of generality that $t_{u_i} + d_{u_i v} < t_{u_j} + d_{u_j v}$ for $j > i$. Here, the firing times $t_{u_i}$, $i = 2, \ldots, n$, are suitable constants. We will show that even in this extended setting $t_v \neq f_i$ still holds and thereby also the claim.

For the sake of contradiction, assume that $t_v(x) = f_1(x)$ for all $x \in [a, b]$. This implies that there exists an index set $J \subset \{1, \ldots, n\}$ with $\sum_{j \in J} w_{u_j v} > 0$ and a corresponding interval $(a_1, 0] \subset [a, b]$ such that

$$c_1 = t_v(x) = \frac{1}{\sum_{i \in J} w_{u_i v}} \left( \theta_v + \sum_{i \in J} w_{u_i v}(t_{u_i} + d_{u_i v}) \right) \quad \text{for all } x \in (a_1, 0],$$

where we have applied (7). Moreover, $J$ is of the form $J = \{2, \ldots, \ell\}$ for some $\ell \in \{1, \ldots, n\}$ because $(t_{u_i} + d_{u_i v})_{i=2}^{n}$ is in ascending order, i.e., if the spike from $u_\ell$ has reached $v$ before $v$ fired, then so did the spikes from $u_i$, $2 \leq i < \ell$. Additionally, we know that $1 \notin J$ since otherwise $t_v$ is non-constant on $(a_1, 0]$ (due to the contribution from $u_1$), i.e., $t_v \neq c_1$ on $(a_1, 0]$. In particular, the spike from $u_1$ reaches $v$ after the neurons $u_2, \ldots, u_\ell$ already caused $v$ to fire, i.e., we have

$$x + d_{u_1 v} \geq t_v(x) = c_1 \quad \text{for all } x \in (a_1, 0].$$

However, it immediately follows that

$$x + d_{u_1 v} > d_{u_1 v} \geq c_1 \quad \text{for all } x > 0.$$

Thus, we obtain $t_v(x) = c_1$ for $x > 0$ (since the spike from $u_1$ still reaches $v$ only after $v$ emitted a spike), which contradicts $t_v(x) = f_1(x)$ for all $x \in [a, b]$.

We perform a similar analysis to show that $f_2$ can not be emulated. For the sake of contradiction, assume that $t_v(x) = f_2(x)$ for all $x \in [a, b]$. This implies that there exists an index set $I \subset \{1, \ldots, n\}$ with $\sum_{i \in I} w_{u_i v} > 0$ and a corresponding interval $(a_2, 0) \subset [a, b]$ such that

$$x + c_1 = t_v(x) = \frac{1}{\sum_{i \in I} w_{u_i v}} \left( \theta_v + w_{u_1 v}(x + d_{u_1 v}) + \sum_{i \in I \setminus \{1\}} w_{u_i v}(t_{u_i} + d_{u_i v}) \right) \quad \text{for } x \in (a_2, 0), \tag{18}$$

where we have applied (7). We immediately observe that $1 \in I$, since otherwise $t_v$ is constant on $(a_2, 0)$. Moreover, by the same reasoning as before we can write $I = \{1, \ldots, \ell\}$ for some $\ell \in \{1, \ldots, n\}$. In order for $t_v(x) = f_2(x)$ for all $x \in [a, b]$ to hold, there needs to exist an index set $J \subset \{1, \ldots, n\}$ with $\sum_{j \in J} w_{u_j v} > 0$ and a corresponding interval $[0, b_2) \subset [a, b]$ such that $t_v = c_1$ on $[0, b_2)$. We conclude that $J = \{1, \ldots, m\}$ or $J = \{2, \ldots, m\}$ for some $m \in \{1, \ldots, n\}$. In the former case, $t_v$ is non-constant on $[0, b_2)$ (due to the contribution from $u_1$), i.e., $t_v \neq c_1$ on $[0, b_2)$. Hence, it remains to consider the latter case. Note that $m < \ell$ implies that $b_2 \leq a_2$ (as $u_2, \ldots, u_m$ already triggered a firing of $v$ before the spike from $u_\ell$ arrived) contradicting the construction $a_2 < 0 < b_2$. Similarly, $m = \ell$, i.e., $J = I \setminus \{1\}$ is not valid because (18) requires that

$$\frac{w_{u_1 v}}{\sum_{i \in I} w_{u_i v}} = 1 \Leftrightarrow \sum_{i \in I \setminus \{1\}} w_{u_i v} = 0 \Leftrightarrow \sum_{j \in J} w_{u_j v} = 0.$$

Finally, $m > \ell$ also results in a contradiction since

$$0 < \sum_{j \in J} w_{u_j v} = \sum_{i \in I \setminus \{1\}} w_{u_i v} + \sum_{j \in J \setminus I} w_{u_j v} = \sum_{j \in J \setminus I} w_{u_j v}$$

together with

$$0 < \sum_{i \in I} w_{u_i v} = \sum_{i \in I \setminus \{1\}} w_{u_i v} + w_{u_1 v} = w_{u_1 v}$$

imply that the neurons $\{u_j : j \in \{1\} \cup J\}$ also trigger a spike in $v$. However, the corresponding interval where the firing of $v$ is caused by $\{u_j : j \in \{1\} \cup J\}$ is necessarily located between $(a_2, 0)$ and $[0, b_2)$, which is not possible. $\square$

**Remark 13.** *The proof shows that $-f_1$ also can not be emulated by a one-layer LSRM. Moreover, by adjusting (18) we observe that a necessary condition for $-f_2$ to be realized is that*

$$\frac{w_{u_1 v}}{\sum_{i \in I} w_{u_i v}} = -1 \Leftrightarrow -\sum_{i \in I \setminus \{1\}} w_{u_i v} = 2w_{u_1 v} \Leftrightarrow -\frac{1}{2} \sum_{i \in I \setminus \{1\}} w_{u_i v} = w_{u_1 v}.$$

*Under this condition $-f_2$ can indeed be realized by a one-layer LSRM as the following statement confirms.*

**Proposition 3.** *Let $a < 0 < b, c$ and consider $f : [a, b] \to \mathbb{R}$ defined as*

$$f(x) = \begin{cases} -x + c & , \text{ if } x < 0 \\ c & , \text{ if } x \geq 0 \end{cases}.$$

*There exists a one-layer LSRM $\Phi$ with output neuron $v$ and input neuron $u_1$ such that $t_v(x) = f(x)$ on $[a, b]$, where $t_v(x)$ denotes the firing time of $v$ on input $t_{u_1} = x$.*

*Proof.* We introduce an auxiliary input neuron with constant firing time $t_{u_2} \in \mathbb{R}$ and specify the parameter of $\Phi = ((W, D, \Theta))$ in the following manner (see Figure 3a):

$$W = \begin{pmatrix} -\frac{1}{2} \\ 1 \end{pmatrix}, D = \begin{pmatrix} d_1 \\ d_2 \end{pmatrix}, \Theta = \theta,$$

where $\theta, d_1, d_2 > 0$ are to be specified. Note that either $u_2$ or $u_1$ together with $u_2$ can trigger a spike in $v$ since $w_{u_1 v} < 0$. Therefore, applying (7) yields that $u_2$ triggers a spike in $v$ under the following circumstances:

$$t_v(x) = \theta + t_{u_2} + d_2 \quad \text{if } t_v(x) \le t_{u_1} + d_1 = x + d_1.$$

Hence, this case only arises when

$$\theta + t_{u_2} + d_2 \le x + d_1 \Leftrightarrow \theta + t_{u_2} + d_2 - d_1 \le x.$$

To emulate $f$ the parameter needs to satisfy

$$\theta + t_{u_2} + d_2 - d_1 \le x \text{ for all } x \in [0, b] \quad \text{and} \quad \theta + t_{u_2} + d_2 - d_1 > x \text{ for all } x \in [a, 0)$$

which simplifies to

$$\theta + t_{u_2} + d_2 - d_1 = 0. \tag{19}$$

If the additional condition

$$\theta + t_{u_2} + d_2 = c \tag{20}$$

is met, we can infer that

$$t_v(x) = \begin{cases} 2(\theta + t_{u_2} + d_2) - (x + d_1) & \text{, if } x < 0 \\ \theta + t_{u_2} + d_2 & \text{, if } x \ge 0 \end{cases} = \begin{cases} -x + c & \text{, if } x < 0 \\ c & \text{, if } x \ge 0 \end{cases}.$$

Finally, it is immediate to verify that the conditions (19) and (20) can be satisfied simultaneously due to the assumption that $c > 0$, e.g., choosing $d_1 = d_2 = c$ and $t_{u_2} = -\theta$ is sufficient. □

**Remark 14.** *We wish to mention that we can not adapt the previous construction to emulate ReLU with a consistent encoding scheme, i.e., such that the input and output firing times encode analog values in the same format with respect to reference times $T_{in}, T_{out} \in \mathbb{R}$, $T_{in} < T_{out}$. Indeed, it is obvious that using the input encoding $T_{in} + x$ and output decoding $-T_{out} + t_v$, does not realize ReLU. Similarly, one verifies that the input encoding $T_{in} - x$ and output decoding $T_{out} - t_v$ also does not yield the desired function. However, choosing the input encoding $T_{in} - x$ and output decoding $-T_{out} + t_v$ gives*

$$\mathcal{R}_\Phi(x) = \begin{cases} -T_{out} - T_{in} + c + x & \text{, if } x > T_{in} \\ -T_{out} + c & \text{, if } x \le T_{in} \end{cases}.$$

*Setting $T_{in} = 0$ and $T_{out} = c$ implies that $\Phi$ realizes ReLU with inconsistent encoding $T_{in} - x$ and $T_{out} + \mathcal{R}_\Phi(x)$. Nevertheless, we want a consistent encoding scheme that allows us to compose ReLU (as typically is the case in ANNs) whereby an inconsistent scheme is disadvantageous.*

Applying the previous construction and adding another layer is adequate to emulate $f_1$ defined in Proposition 2 by a two-layer LSRM.

**Proposition 4.** *Let $a < 0 < b < 0.5 \cdot c$ and consider $f : [a, b] \to \mathbb{R}$ defined as*

$$f(x) = \begin{cases} x + c & \text{, if } x > 0 \\ c & \text{, if } x \le 0 \end{cases}$$

*There exists a two-layer LSRM $\Phi$ with output neuron $v$ and input neuron $u_1$ such that $t_v(x) = f(x)$ on $[a, b]$, where $t_v(x)$ denotes the firing time of $v$ on input $t_{u_1} = x$.*

*Proof.* We introduce an auxiliary input neuron $u_2$ with constant firing time $t_{u_2} \in \mathbb{R}$ and specify the parameter of $\Phi = ((W^1, D^1, \Theta^1), (W^2, D^2, \Theta^2))$ in the following manner:

$$W^1 = \begin{pmatrix} -\frac{1}{2} & 0 \\ 1 & 2 \end{pmatrix}, D^1 = \begin{pmatrix} d & 0 \\ d & \frac{d}{2} \end{pmatrix}, \Theta^1 = \begin{pmatrix} \theta \\ 2\theta \end{pmatrix}, W^2 = \begin{pmatrix} -\frac{1}{2} \\ 1 \end{pmatrix}, D^2 = \begin{pmatrix} d \\ d \end{pmatrix}, \Theta^2 = \theta, \tag{21}$$

where $d \ge 0$ and $\theta > 0$ is chosen such that $\theta + t_{u_2} > b$. We denote the input neurons by $u_1, u_2$, the neurons in the hidden layer by $z_1, z_2$ and the output neuron by $v$. Note that the firing time of

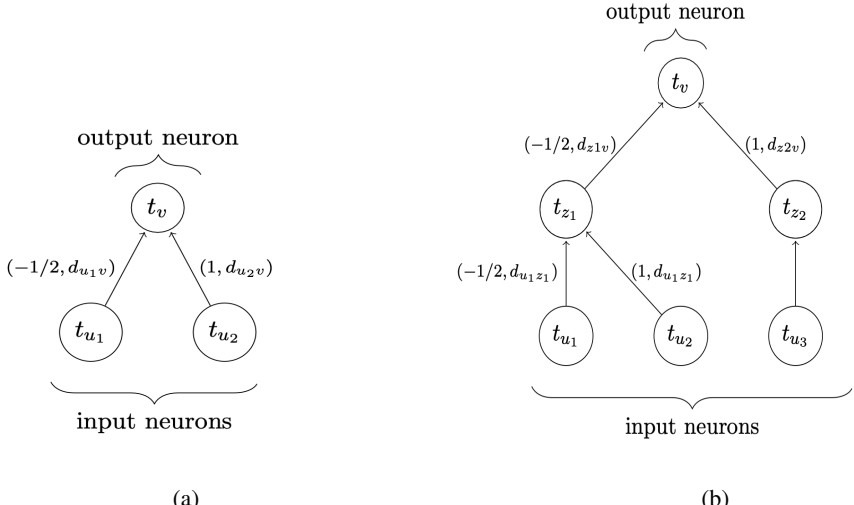

Figure 3: (a) Computation graph associated with an LSRM with two input neurons and one output neuron that realizes $f$ as defined in Proposition 3. (b) Stacking the network in (a) twice results in an LSRM that realizes the ReLU activation function.

$z_1$ depends on $u_1$ and $u_2$. In particular, either $u_2$ or $u_1$ together with $u_2$ can trigger a spike in $z_1$ since $w_{u_1 z_1} < 0$. Therefore, applying (7) yields that $u_2$ triggers a spike in $z_1$ under the following circumstances:

$$t_{z_1}(x) = \theta + t_{u_2} + d \quad \text{if } t_{z_1}(x) \leq t_{u_1} + d = x + d.$$

Hence, this case only arises when

$$\theta + t_{u_2} + d \leq x + d \Leftrightarrow \theta + t_{u_2} \leq x. \tag{22}$$

However, by construction $\theta + t_{u_2} > b$, so that (22) does not hold for any $x \in [a, b]$. Thus, we conclude via (7) that

$$t_{z_1}(x) = 2(\theta + t_{u_2} + d) - (x + d) = 2(\theta + t_{u_2}) + d - x.$$

By construction, the firing time $t_{z_2} = \theta + 2t_{u_2} + d$ of $z_2$ is a constant which depends on the input only via $u_2$. A similar analysis as in the first layer shows that

$$t_v(x) = \theta + t_{z_2} + d \quad \text{if } t_v(x) \leq t_{z_1} + d = 2(\theta + t_{u_2}) + d - x + d = 2(\theta + t_{u_2} + d) - x.$$

Hence, $z_2$ triggers a spike in $v$ when

$$\theta + \theta + 2t_{u_2} + d + d \leq 2(\theta + t_{u_2} + d) - x \quad \Leftrightarrow \quad x \leq 0.$$

If the additional condition

$$\theta + t_{z_2} + d = c \quad \Leftrightarrow \quad 2(\theta + d + t_{u_2}) = c \tag{23}$$

is met, we can infer that

$$
t_v(x) =
\begin{cases}
2(\theta + t_{z_2} + d) - (t_{z_1}(x) + d) & \text{, if } x > 0 \\
\theta + t_{z_2} + d & \text{, if } x \leq 0
\end{cases}
$$
$$
=
\begin{cases}
2c - (2(\theta + t_{u_2}) + d - x + d) & \text{, if } x > 0 \\
c & \text{, if } x \leq 0
\end{cases}
$$
$$
=
\begin{cases}
x + c & \text{, if } x > 0 \\
c & \text{, if } x \leq 0
\end{cases}.
$$

Choosing $\theta$, $t_{u_2}$ and $d$ sufficiently small under the given constraints guarantees that (23) holds, i.e., $\Phi$ emulates $f$ as desired. $\qquad\square$

**Remark 15.** *It is again important to specify the encoding scheme via reference times $T_{in}, T_{out} \in \mathbb{R}$, $T_{in} < T_{out}$ to ensure that $\Phi$ realizes ReLU. The input encoding $T_{in} - x$ and output decoding $T_{out} - t_v$ does not yield the desired output since it results in a realization of the type $-ReLU(-x)$. In contrast, the input encoding $T_{in} + x$ and output decoding $-T_{out} + t_v$ with $T_{in} = 0$ and $T_{out} = c$ gives*

$$\mathcal{R}_\Phi(x) = -T_{out} + t_v(T_{in} + x) = -T_{out} + f(T_{in} + x) = \begin{cases} x & , \text{if } x > 0 \\ 0 & , \text{if } x \leq 0 \end{cases} = ReLU(x).$$

*In this case, it is necessary to choose the reference time $T_{in} = 0$ to ensure that the breakpoint is also at zero. Next, we show that there is actually more freedom in choosing the reference time by analysing the construction in the proof more carefully.*

**Proposition 5.** *Let $a < 0 < b$ and consider $f : [a, b] \to \mathbb{R}$ defined as*

$$f(x) = \begin{cases} x & , \text{if } x > 0 \\ 0 & , \text{if } x \leq 0 \end{cases}$$

*There exists a two-layer LSRM $\Phi$ with realization $\mathcal{R}_\Phi = f$ on $[a, b]$ with encoding scheme $T_{in} + x$ and decoding $-T_{out} + t_v$, where $v$ is the output neuron of $\Phi$, $T_{in} \in \mathbb{R}$ and $T_{out} = T_{in} + c$ for some constant $c > 0$ depending on the parameters of $\Phi$.*

*Proof.* Performing a similar construction and the same analysis as in the proof of Proposition 4 yields the claim. First, we slightly adjust $\Phi = ((W^1, D^1, \Theta^1), (W^2, D^2, \Theta^2))$ in comparison to (21) and consider the network

$$W^1 = \begin{pmatrix} -\frac{1}{2} & 0 \\ 1 & 1 \end{pmatrix}, D^1 = \begin{pmatrix} d & 0 \\ d & d \end{pmatrix}, \Theta^1 = \begin{pmatrix} \theta \\ \theta \end{pmatrix}, W^2 = \begin{pmatrix} -\frac{1}{2} \\ 1 \end{pmatrix}, D^2 = \begin{pmatrix} d \\ d \end{pmatrix}, \Theta^2 = \theta,$$

where $d \geq 0$ and $\theta > b$ are fixed (see Figure 3b). Second, we choose the input reference time $T_{in} \in \mathbb{R}$ and fix the input of the auxiliary input neuron $u_2$ as $t_{u_2} = T_{in} \in \mathbb{R}$. Finally, setting the output reference time $T_{out} = 2(\theta + d) + T_{in}$ is sufficient to guarantee that $\Phi$ realizes $f$ on $[a, b]$. □

### A.4 REALIZING ReLU NETWORKS BY SPIKING NEURAL NETWORKS

In this section, we show that an LSRM has the capability to reproduce the output of any ReLU network. Specifically, given access to the weights and biases of an ANN, we construct an LSRM and set the parameter values based on the weights and biases of the given ANN. This leads us to the desired result. The essential part of our proof revolves around choosing the parameters of an LSRM such that it effectively realizes the composition of an affine-linear map and the non-linearity represented by the ReLU activation. The realization of ReLU with LSRMs is proved in the previous Section A.3. To realize an affine-linear function using a LSRM neuron, it is necessary to ensure that the spikes from all the input neurons together result in the firing of an output neuron instead of any subset of the input neurons. We achieve that by appropriately adjusting the value of the threshold parameter. As a result, a LSRM neuron, which implements an affine-linear map, avoids partitioning of the input space.

**Setup for the proof of Theorem 3** Let $d, L \in \mathbb{N}$ be the width and the depth of an ANN $\Psi$, respectively, i.e.,

$$\Psi = ((A^1, B^1), (A^2, B^2), \dots, (A^L, B^L)), \text{ where } (A^\ell, B^\ell) \in \mathbb{R}^{d \times d} \times \mathbb{R}^d, 1 \leq \ell < L,$$
$$(A^L, B^L) \in \mathbb{R}^{1 \times d} \times \mathbb{R}.$$

For a given input domain $[a, b]^d \subset \mathbb{R}^d$, we denote by $\Psi^\ell = ((A^\ell, B^\ell))$ the $\ell$-th layer, where $y^0 \in [a, b]^d$ and

$$y^l = \mathcal{R}_{\Psi^l}(y^{l-1}) = \sigma(A^l y^{l-1} + B^l), 1 \leq \ell < L,$$
$$y^L = \mathcal{R}_{\Psi^L}(y^{L-1}) = A^L y^{L-1} + B^L \tag{24}$$

so that $\mathcal{R}_\Psi = \mathcal{R}_{\Psi^L} \circ \cdots \circ \mathcal{R}_{\Psi^1}$.

For the construction of the corresponding LSRM we refer to the associated weights and delays between two LSRM neurons $u$ and $v$ by $w_{uv}$ and $d_{uv}$, respectively.

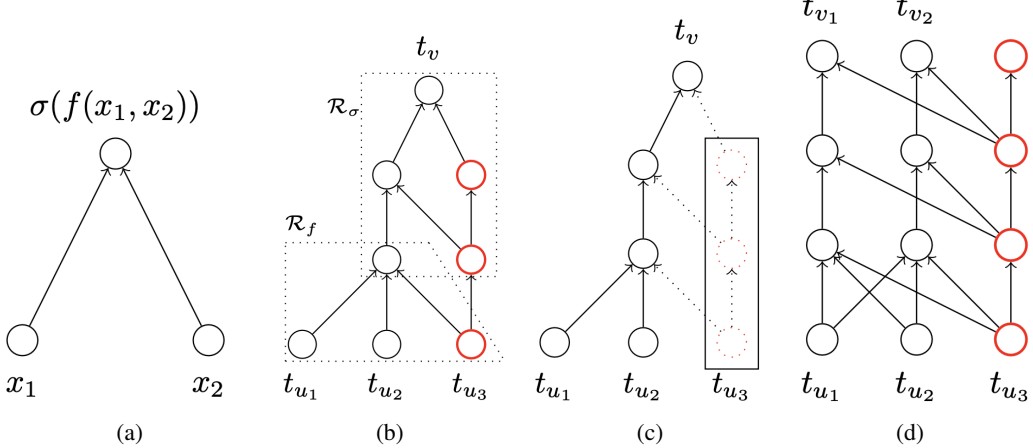

Figure 4: (a) Computation graph of an ANN with two input and one output unit realizing $\sigma(f(x_1, x_2))$, where $\sigma$ is the ReLU activation function. (b) Computation graph associated with an LSRM resulting from the concatenation of $\Phi^\sigma$ and $\Phi^f$ that realizes $\sigma(f(x_1, x_2))$. The auxiliary neurons are shown in red. (c) Same computation graph as in (b); when parallelizing two identical networks, the dotted auxiliary neurons can be removed and auxiliary neurons from (b) can be used for each network instead. (d) Computation graph associated with an LSRM as a result of the parallelization of two subnetworks $\Phi^{\sigma \circ f_1}$ and $\Phi^{\sigma \circ f_2}$. The auxiliary neuron in the output layer serves the same purpose as the auxiliary neuron in the input layer and is needed when concatenating two such subnetworks $\Phi_{\sigma \circ f}$.

***Proof of Theorem 3***. Any multi-layer ANN $\Psi$ with ReLU activation is simply an alternating composition of affine-linear functions $A^l y^{l-1} + B^l$ and a non-linear function represented by $\sigma$. To generate the mapping realized by $\Psi$, it suffices to realize the composition of affine-linear functions and the ReLU non-linearity and then extend the construction to the whole network using concatenation and parallelization operations. We prove the result via the following steps; see also Figure 4 for a depiction of the intermediate constructions.

**Step 1:** Realizing ReLU non-linearity.
Proposition 5 gives the desired result.

**Step 2:** Realizing affine-linear functions with one-dimensional range.
Let $f : [a, b]^d \to \mathbb{R}$ be an affine-linear function

$$f(x) = C^T x + s, \quad C^T = (c_1, \ldots, c_d) \in \mathbb{R}^d, s \in \mathbb{R}. \tag{25}$$

Consider a one-layer LSRM that consists of an output neuron $v$ and d input units $u_1, \ldots, u_d$. Via (7) the firing time of $v$ as a function of the input firing times on the linear region $R^I$ corresponding to the index set $I = \{1, \ldots, d\}$ is given by

$$t_v(t_{u_1}, \ldots, t_{u_d}) = \frac{\theta_v}{\sum_{i \in I} w_{u_i v}} + \frac{\sum_{i \in I} w_{u_i v}(t_{u_i} + d_{u_i v})}{\sum_{i \in I} w_{u_i v}} \quad \text{provided that} \quad \sum_{i \in I} w_{u_i v} > 0.$$

Introducing an auxiliary input neuron $u_{d+1}$ with weight $w_{u_{d+1} v} = 1 - \sum_{i \in I} w_{u_i v}$ ensures that $\sum_{i \in I \cup \{d+1\}} w_{u_i v} > 0$ and leads to the firing time

$$t_v(t_{u_1}, \ldots, t_{u_{d+1}}) = \theta_v + \sum_{i \in I \cup \{d+1\}} w_{u_i v}(t_{u_i} + d_{u_i v}) \quad \text{on } R^{I \cup \{d+1\}}.$$

Setting $w_{u_i v} = c_i$ for $i \in I$ and $d_{u_j v} = d' \geq 0$ for $j \in I \cup \{d+1\}$ yields

$$t_v(t_{u_1}, \ldots, t_{u_{d+1}}) = \theta_v + w_{u_{d+1} v} \cdot t_{u_{d+1}} + d' + \sum_{i \in I} c_i t_{u_i} \text{ on } R^{I \cup \{d+1\}} \cap [a, b]^d.$$

Therefore, an LSRM $\Phi^f = (W, D, \Theta)$ with parameters

$$W = \begin{pmatrix} c_1 \\ \vdots \\ c_{d+1} \end{pmatrix}, D = \begin{pmatrix} d' \\ \vdots \\ d' \end{pmatrix}, \Theta = \theta > 0, \quad \text{where } c_{d+1} = 1 - \sum_{i \in I} c_i,$$

and the usual encoding scheme $T_{\text{in}}/T_{\text{out}} + \cdot$ and fixed firing time $t_{u_{d+1}} = T_{\text{in}} \in \mathbb{R}$ realizes

$$\mathcal{R}_{\Phi^f}(x) = -T_{\text{out}} + t_v(T_{\text{in}} + x_1, \ldots, T_{\text{in}} + x_d, T_{\text{in}}) = -T_{\text{out}} + \theta + T_{\text{in}} + d' + \sum_{i \in I} c_i x_i \quad (26)$$

$$= -T_{\text{out}} + \theta + T_{\text{in}} + d' + f(x_1, \ldots, x_d) - s \quad \text{on } R^{I \cup \{d+1\}} \cap [a, b]^d. \quad (27)$$

Choosing a large enough threshold $\theta$ ensures that a spike in $v$ is necessarily triggered after all the spikes from $u_1, \ldots, u_{d+1}$ reached $v$ so that $[a, b]^d \subset R^{I \cup \{d+1\}}$ holds. It suffices to set

$$\theta \geq \sup_{x \in [a,b]^d} \sup_{x_{\min} \leq t - T_{\text{in}} - d' \leq x_{\max}} P_v(t),$$

where $x_{\min} = \min\{x_1, \ldots, x_d, 0\}$ and $x_{\max} = \max\{x_1, \ldots, x_d, 0\}$, since this implies that the potential $P_v(t)$ is smaller than the threshold to trigger a spike in $v$ on the time interval associated to feasible input spikes, i.e., $v$ emits a spike after the last spike from an input neuron arrived at $v$. Applying (5) shows that for $x \in [a, b]^d$ and $t \in [x_{\min} + T_{\text{in}} + d', x_{\max} + T_{\text{in}} + d']$

$$P_v(t) = \sum_{i \in I} w_{u_i v}(t - (T_{\text{in}} + x_i) - d_{u_i v}) + w_{u_{d+1} v}(t - T_{\text{in}} - d_{u_{d+1} v}) = t - d' - T_{\text{in}} + \sum_{i \in I} c_i x_i$$

$$\leq x_{\max} + d \|C\|_\infty \|x\|_\infty \leq (1 + d \|C\|_\infty) \max\{|a|, |b|\}.$$

Hence, we set

$$\theta = (1 + d \|C\|_\infty) \max\{|a|, |b|\} + s + |s| \quad \text{and} \quad T_{\text{out}} = \theta - s + T_{\text{in}} + d'$$

to obtain via (26) that

$$\mathcal{R}_{\Phi^f}(x) = -T_{\text{out}} + t_v(T_{\text{in}} + x_1, \ldots, T_{\text{in}} + x_d, T_{\text{in}}) = f(x) \quad \text{for } x \in [a, b]^d. \quad (28)$$

Note that the reference time $T_{\text{out}} = (1 + d \|C\|_\infty) \max\{|a|, |b|\} + |s| + T_{\text{in}} + d'$ is independent of the specific parameters of $f$ in the sense that only upper bounds $\|C\|_\infty, |s|$ on the parameters are relevant. Therefore, $T_{\text{out}}$ (with the associated choice of $\theta$) can be applied for different affine linear functions as long as the upper bounds remain valid. This is necessary for the composition and parallelization of subnetworks in the subsequent construction.

**Step 3:** Realizing compositions of affine-linear functions with one-dimensional range and ReLU. The next step is to realize the composition of ReLU $\sigma$ with an affine linear mapping $f$ defined in (25). To that end, we want to concatenate the networks $\Phi^\sigma$ and $\Phi^f$ constructed in Step 1 and Step 2, respectively, via Lemma 1. To employ the concatenation operation we need to perform the following steps:

1. Find an appropriate input domain $[a', b'] \subset \mathbb{R}$, that contains the image $f([a, b]^d)$ so that parameters and reference times of $\Phi^\sigma$ can be fixed appropriately (see Proposition 5 for the detailed conditions on how to choose the parameter).

2. Ensure that the output reference time $T_{\text{out}}^f$ of $\Phi^f$ equals the input reference time $T_{\text{in}}^\sigma$ of $\Phi^\sigma$.

3. Ensure that the number of neurons in the output layer of $\Phi^f$ is the same as the number of input neurons in $\Phi^\sigma$.

For the first point, note that

$$|f(x)| = |C^T x + s| \leq d \|C\|_\infty \cdot \|x\|_\infty + |s| \leq d \|C\|_\infty \cdot \max\{|a|, |b|\} + |s| \text{ for all } x \in [a, b]^d.$$

Hence, we can use the input domain

$$[a', b'] = [-d \|C\|_\infty \cdot \max\{|a|, |b|\} + |s|, d \|C\|_\infty \cdot \max\{|a|, |b|\} + |s|]$$

and specify the parameters of $\Phi^\sigma$ accordingly. Additionally, recall from Proposition 5 that $T_{\text{in}}^\sigma$ can be chosen freely, so we may fix $T_{\text{in}}^\sigma = T_{\text{out}}^f$, where $T_{\text{out}}^f$ is established in Step 2. It remains to consider the third point. In order to realize ReLU an additional auxiliary neuron in the input layer of $\Phi^\sigma$ with constant input $T_{\text{in}}^\sigma$ was introduced. Hence, we also need to add an additional output neuron in $\Phi^f$ with (constant) firing time $T_{\text{out}}^f = T_{\text{in}}^\sigma$ so that the corresponding output and input dimension and their specification match. This is achieved by introducing a single synapse from the auxiliary neuron in the input layer of $\Phi^f$ to the newly added output neuron and by specifying the parameters of the newly introduced synapse and neuron suitably. Formally, the adapted network $\Phi^f = (W, D, \Theta)$ is given by

$$
W = \begin{pmatrix} c_1 & 0 \\ \vdots & \vdots \\ c_d & 0 \\ c_{d+1} & 1 \end{pmatrix}, D = \begin{pmatrix} d' & 0 \\ \vdots & \vdots \\ d' & 0 \\ d' & d' \end{pmatrix}, \Theta = \begin{pmatrix} \theta \\ T_{\text{out}}^f - T_{\text{in}}^f - d' \end{pmatrix},
$$

where the values of the parameters are specified in Step 2.

Then the realization of the concatenated network $\Phi^{\sigma \circ f}$ is the composition of the individual realizations. This is exemplarily demonstrated in Figure 4b for the two-dimensional input case. By analyzing $\Phi^{\sigma \circ f}$, we conclude that a three-layer LSRM with

$$
N(\Phi^{\sigma \circ f}) = N(\Phi^\sigma) - N_0(\Phi^\sigma) + N(\Phi^f) = 5 - 2 + d + 3 = d + 6
$$

computational units can realize $\sigma \circ f$ on $[a, b]^d$, where $N_0(\Phi^\sigma)$ denotes the number of neurons in the input layer of $\Phi^\sigma$.

**Step 4:** Realizing layer-wise computation of $\Psi$.
The computations performed in a layer $\Psi^\ell$ of $\Psi$ are described in (8). Hence, for $1 \leq \ell < L$ the computation can be expressed as

$$
\mathcal{R}_{\Psi^\ell}(y^{l-1}) = \sigma(A^l y^{l-1} + B^l) = \begin{pmatrix} \sigma(\sum_{i=1}^d A_{1,i}^l y_i^{l-1} + B_1^l) \\ \vdots \\ \sigma(\sum_{i=1}^d A_{d,i}^l y_i^{l-1} + B_d^l) \end{pmatrix} =: \begin{pmatrix} \sigma(f_1(y^{l-1})) \\ \vdots \\ \sigma(f_d(y^{l-1})) \end{pmatrix},
$$

where $f_1^\ell, \ldots, f_d^\ell$ are affine linear functions with one-dimensional range on the same input domain $[a^{\ell-1}, b^{\ell-1}] \subset \mathbb{R}^d$, where $[a^0, b^0] = [a, b]$ and $[a^\ell, b^\ell]$ is the range of

$$
(\sigma \circ f_1^{\ell-1}, \ldots, \sigma \circ f_d^{\ell-1})([a^{\ell-1}, b^{\ell-1}]^d).
$$

Thus, via Step 3, we construct LSRMs $\Phi_1^\ell, \ldots, \Phi_d^\ell$ that realize $\sigma \circ f_1^\ell, \ldots, \sigma \circ f_d^\ell$ on $[a^{\ell-1}, b^{\ell-1}]$. Note that by choosing appropriate parameters in the construction performed in Step 2 (as described below (28)), e.g., $\|A^l\|_\infty$ and $\|B^l\|_\infty$, we can employ the same input and output reference time for each $\Phi_1^\ell, \ldots, \Phi_d^\ell$. Consequently, we can parallelize $\Phi_1^\ell, \ldots, \Phi_d^\ell$ (see Lemma 2) and obtain networks $\Phi^\ell = P(\Phi_1^\ell, \ldots, \Phi_d^\ell)$ realizing $\mathcal{R}_{\Psi^\ell}$ on $[a^{\ell-1}, b^{\ell-1}]$. Finally, $\Psi^L$ can be directly realized via Step 2 by an LSRM $\Phi^L$ (as in the last layer no activation function is applied and the output is one-dimensional). Although $\Phi^\ell$ already performs the desired task of realizing $\mathcal{R}_{\Psi^\ell}$ we can slightly simplify the network. By construction in Step 3, each $\Phi_i^\ell$ contains two auxiliary neurons in the hidden layers. Since the input and output reference time is chosen consistently for $\Phi_1^\ell, \ldots, \Phi_d^\ell$, we observe that the auxiliary neurons in each $\Phi_i^\ell$ perform the same operations and have the same firing times. Therefore, without changing the realization of $\Phi^\ell$ we can remove the auxiliary neurons in $\Phi_2^\ell, \ldots, \Phi_d^\ell$ and introduce synapses from the auxiliary neurons in $\Phi_1^\ell$ accordingly. This is exemplarily demonstrated in Figure 4c for the case $d = 2$. After this modification, we observe that $L(\Phi^\ell) = L(\Phi_i^\ell) = 3$ and

$$
N(\Phi^\ell) = N(\Phi_1^\ell) + \sum_{i=2}^d \left( N(\Phi_i^\ell) - 2 - N_0(\Phi_i^\ell) \right) = dN(\Phi_1^\ell) - (d-1)(2 + N_0(\Phi_1^\ell))
$$

$$
= d(d+6) - 2(d-1) - (d-1)(d+1) = 4d + 3 \quad \text{for } 1 \leq \ell < L,
$$

whereas $L(\Phi^L) = 1$ and $N(\Phi^L) = d + 2$.

**Step 5:** Realizing compositions of layer-wise computations of $\Psi$.

The last step is to compose the realizations $\mathcal{R}_{\Phi^1}, \ldots, \mathcal{R}_{\Phi^L}$ to obtain the realization

$$\mathcal{R}_{\Phi^L} \circ \cdots \circ \mathcal{R}_{\Phi^1} = \mathcal{R}_{\Psi^L} \circ \cdots \circ \mathcal{R}_{\Psi^1} = \mathcal{R}_{\Psi}.$$

As in Step 3, it suffices again to verify that the concatenation of the networks $\mathcal{R}_{\Phi^1}, \ldots, \mathcal{R}_{\Phi^L}$ is feasible. First, note that for $\ell = 1, \ldots, L$ the input domain of $\mathcal{R}_{\Phi^\ell}$ is given by $[a^{\ell-1}, b^{\ell-1}]$ so that, we can fix the suitable output reference time $T_{\text{out}}^{\Phi^\ell}$ based on the parameters of the network, the input domain $[a^{\ell-1}, b^{\ell-1}]$, and some input reference time $T_{\text{in}}^{\Phi^\ell} \in \mathbb{R}$. By construction in Steps 2 - 4 $T_{\text{in}}^{\Phi^\ell}$ can be chosen freely. Hence setting $T_{\text{in}}^{\Phi^{\ell+1}} = T_{\text{out}}^{\Phi^\ell}$ ensures that the reference times of the corresponding networks agree. It is left to align the input dimension of $\Phi^{\ell+1}$ and the output dimension of $\Phi^\ell$ for $\ell = 1, \ldots, L-1$. Due to the auxiliary neuron in the input layer of $\Phi^{\ell+1}$, we also need to introduce an auxiliary neuron in the output layer of $\Phi^\ell$ (see Figure 4d) with the required firing time $T_{\text{in}}^{\Phi^{\ell+1}} = T_{\text{out}}^{\Phi^\ell}$. Similarly, as in Step 3, it suffices to add a single synapse from the auxiliary neuron in the previous layer to obtain the desired firing time.

Thus, we conclude that $\Phi = \Phi^L \bullet \cdots \bullet \Phi^1$ realizes $\mathcal{R}_{\Psi}$ on $[a, b]$, as desired. The complexity of $\Phi$ in the number of layers and neurons is given by

$$L(\Phi) = \sum_{\ell=1}^{L} L(\Phi^\ell) = 3L - 2 = 3L(\Psi) - 2$$

and

$$\begin{aligned}
N(\Phi) &= N(\Phi^1) + \sum_{\ell=2}^{L} \left( N(\Phi^\ell) - N_0(\Phi^\ell) \right) + (L-1) \\
&= 4d + 3 + (L-2)(4d + 3 - (d+1)) + (d + 2 - (d+1)) + (L-1) \\
&= 3L(d+1) - (2d+1) \\
&= N(\Psi) + L(2d+3) - (2d+2)
\end{aligned}$$

$\square$

**Remark 16.** *Note that the delays play no significant role in the proof of the above theorem. Nevertheless, they can be employed to alter the timing of spikes, consequently impacting the firing time and the resulting output. However, the exact function of delays requires further investigation. The primary objective is to present a construction that proves the existence of an LSRM capable of accurately reproducing the output of any ReLU network.*

A.5   PROOF OF EXAMPLE 1

Recall the function introduced in Example 1,

$$f(x) = \begin{cases} x, & x \leq -\theta \\ \frac{x-\theta}{2}, & -\theta < x < \theta \\ 0, & x \geq \theta \end{cases} = -\frac{1}{2}\sigma(-x-\theta) - \frac{1}{2}\sigma(-x+\theta) \quad \text{for } x \in [a,b] \subset \mathbb{R}, \theta > 0,$$

that provides insights under which circumstances LSRMs may express some target function with lower complexity than any corresponding ReLU-ANN.

***Proof of Example 1.*** First, we realize $f$ using LSRM neurons. Consider an LSRM $\Phi = (W, D, \Theta)$,

$$W = \begin{pmatrix} 1 \\ 1 \end{pmatrix}, D = \begin{pmatrix} d \\ d \end{pmatrix}, \Theta = \theta$$

where $d \geq 0$. Denote the input neurons by $u_1$ and $u_2$ and the output neuron by $v$. We set the input and output reference times to $T_{\text{in}} = a$ and $T_{\text{out}} = \theta + T_{\text{in}} + d$, respectively. Following the usual encoding scheme, the input neuron $u_1$ fires at time $t_{u_1} = T_{\text{in}} + x$ and we fix $t_{u_2} = T_{\text{in}}$. Next, we consider two cases $(1)\ |x| \geq \theta$ and $(2)\ |x| < \theta$.

**Case (1)** In this case, an isolated spike from either one of the two input neurons causes the output neuron $v$ to fire at time $t_v$. In particular, if $x$ is negative, then $u_1$ triggers the spike in $v$, whereas,

if $x$ is positive, then $u_2$ triggers the spike in $v$. Using equation (7), it can be directly verified that $t_v = T_{\text{out}} + f(x)$, i.e, $\mathcal{R}_\Phi(x) = f(x)$ for $|x| \geq \theta$.

**Case (2)** In this case, a spike in $v$ will be triggered by both the input neurons. By using equation (7), we observe that $t_v = T_{\text{out}} + \frac{x-\theta}{2}$, i.e, $\mathcal{R}_\Phi(x) = f(x)$ for $|x| < \theta$.

Next, observe that $f$ can be realized using a two-layer ReLU-ANN $\Psi = ((A^1, B^1), (A^2, B^2))$ with two units in the hidden layer and one output unit, where

$$A^1 = \begin{pmatrix} -1 & -1 \end{pmatrix}, \quad B^1 = \begin{pmatrix} -\theta & \theta \end{pmatrix}, \quad A^2 = \begin{pmatrix} -0.5 \\ -0.5 \end{pmatrix}, \quad B^2 = 0.$$

Moreover, note that $\Psi$ is optimal in terms of complexity. Indeed, a single ReLU unit with any number of incoming edges separates the input space into at most two linear regions. Since $f$ has three different linear regions, it is not possible to capture all the linear regions by a single ReLU unit. Hence, at least two hidden units, i.e., four units with input and output, and two layers are needed realize $f$ via a ReLU-ANN. $\qquad\square$

