# OpenReview forum: "Are Spiking Neural Networks more expressive than Artificial Neural Networks?"
_ICLR.cc/2024/Conference — Submitted to ICLR 2024_

### Official Review · Reviewer_Azjn · 2023-10-19

**Soundness:** 3 good
**Presentation:** 3 good
**Contribution:** 3 good
**Rating:** 6
**Confidence:** 4

**Summary:**

This paper investigates the approximation capability of a certain type of spiking neural network (the noise-free version of the spike response model with a linear response function) from the perspective of continuous piecewise linear (CPWL) functions. The following conclusions are proved.

1. A spiking neural network (SNN) is a CPWL function under certain encoding schemes and assumptions about synaptic weights (Theorem 1).

2. Both ReLU artificial neural networks (ANN) and SNN can represent certain functions more efficiently (Theorem 2 and Example 1).

3. Any ReLU ANN is equivalent to an SNN, and the size of the SNN is similar to that of ANN (Theorem 3).

4. The maximum number of linear regions that a one-layer SNN generates scales exponentially with the input dimension (constant for one-layer ReLU ANN) (Theorem 4).

**Strengths:**

1. To the best of my knowledge, this is the first theoretical paper that studies the number of linear regions for SNNs. This is an important aspect of understanding the expressive capacity of neural networks and has been widely investigated for traditional ANNs.

2. The basic definitions of SNN used in this paper are listed and explained in detail.

3. This paper provides a thorough comparison between the expressive capacity of SNN and ANN. Theorem 2 and Example 1 are two intuitive examples to understand the difference between SNN and ANN.

**Weaknesses:**

1. The theoretical conclusions are great but it seems that they are highly dependent on the realization of SNN. In Theorem 1, it is assumed that the sum of synaptic weights of each neuron is positive. Later in Remark 3, the authors claim that when the assumption is not met, the realization of the network is not well-defined. This seems to be a strong assumption to me and I am wondering how can we guarantee the positive sum of synaptic weights. In practice, is this assumption met in experiments? What will happen in practice if this assumption is not satisfied?

2. The paper is self-consistent and rigorous but might be hard to follow for common readers. In Section 2, there are many formal definitions, which are general but may lose readability. For example, I think it would be better to provide a concrete formulation (or example) to introduce SNN and leave the formal definition in the appendix. The formal definition using a graph is general but might be hard to understand for most readers.

**Questions:**

1. To my understanding, Theorem 2 can serve as a comparison between SNN and ANN and implies that ANN can express the ReLU function more efficiently. Then I think it would be better to put Theorem 2 in section 3.2, where Theorem 2 and Example 1 study the advantages of ANN and SNN, respectively.

2. The title directly indicates the central concern of this paper but the answer seems to be incomplete. To my understanding, Theorem 3 implies that SNN is at least as expressive as ANN from the perspective of equivalence of approximation. Then a natural question is can SNN be more expressive than ANN, i.e., how many parameters are needed for an ANN to express an SNN? If we can get a similar result as Theorem 3, then we can conclude that ANN and SNN have similar expressive power, otherwise, a separation result confirms that SNN can be more expressive than ANN. Although this paper provides several results (Example 1 and Theorem 4) to demonstrate the advantage of SNN, Example 1 is not a separation result, and Theorem 4 is not demonstrated through the number of parameters. It seems that Theorem 4 can be translated into a separation result to make Theorem 3 more complete. For more literature about separation results, it might be helpful to read [1,2,3].

3. The abbreviation CPWL occurs on page 3 (the 4th line of the 2nd contribution) for the first time, but its full name is introduced at the beginning of section 3.

[1] The power of depth for feedforward neural networks, COLT 2016.

[2] Benefits of depth in neural networks, COLT 2016.

[3] Theoretical exploration of flexible transmitter model, TNNLS 2023.

---

> ### Author Response · Authors · 2023-11-15
>
> We thank the reviewer for the time and effort dedicated to the reviewing of our submitted manuscript. Below, we provide a detailed response organized under the same headings as those used in the review.
>
> ## Weaknesses
> * We acknowledge the reviewer’s assessment that the results are dependent on the realization of SNN. If the sum of weights of a neuron is negative, then a neuron remains inactive and fails to encode any information. The assumption that the sum of synaptic weights of each neuron is positive is a sufficient but not necessary condition. In particular, it guarantees the well-definedness of the input-output function on the whole input domain, i.e., for each input the output neuron will fire (which is not necessarily required in practical applications). However, in each layer, there should be at least one neuron which satisfies the positivity condition to make sure that the information is propagated in the forward layers. In practical scenarios, the issue can be mitigated, e.g., by introducing some weight bumping mechanism as in [1]. They introduce a hyperparameter (a threshold value) which controls the number of neurons without any output spike. If the number of non-spiking neurons is above a certain threshold, then the input weights of the silent neurons are increased, thereby boosting the layer with insufficient spikes. The threshold in the output layer can be set such that all the neurons in the output layer always emit a spike. A similar mechanism can also be applied in our case. We are actively working on the practical implementation and it is currently in progress.
>
> [1] Göltz, J., Kriener, L., Baumbach, A. et al. Fast and energy-efficient neuromorphic deep learning with first-spike times. Nat Mach Intell 3, 823–835 (2021).
>
>
> * We appreciate the reviewer’s concern regarding the potential difficulty for common readers to follow the formal definitions presented in Section 2. While we acknowledge the suggestion to place these details in the appendix for improved readability, we believe that it is important to keep these definitions in the main text for a comprehensive understanding of the technical details throughout the paper. We want to point out that the understanding of the computations involving firing time and terminology associated with spiking neurons is essential for fully grasping the significance of our results.
>
>
> ## Questions
> * In response to the reviewer’s suggestion, we have incorporated Theorem 2 into Section 3.2.
>
> * The question raised by the reviewer, “Can SNN be more expressive than ANN, i.e. how many parameters are needed for an ANN to express an SNN'' is indeed an important one. Theorems 1 and 3 establish that SNNs are at least as expressive as ReLU ANNs, both generating CPWL mappings. While ReLU ANNs can reproduce the output of any SNN, the bound on the number of parameters, neurons or layers required for this equivalence is an open question. The absence of a separation result in Example1 is acknowledged and the reviewer’s observation about Theorem 4 being translated into a separation result is what we plan to do. Understanding the bound on the number of linear regions in more complex settings (eg. multi-layer case) is pivotal, forming the groundwork for subsequent investigations. We also thank the reviewer for suggesting the literature on separation results. As for the title, we acknowledge that the answer is incomplete. It is essential to emphasize that our central focus is on the fundamental question of equivalence and demonstrating the differences in the structure of computations between SNNs and ANNs. While the prospect of SNNs being advantageous over ANNs is intriguing, our current findings represent an important initial step in that direction. It also provides a foundation for future explorations into more complex settings, for instance, incorporating multi-spikes and the coupled effect of response and threshold functions on the approximation results.
>
> >  The abbreviation CPWL occurs on page 3 (the 4th line of the 2nd contribution) for the first time, but its full name is introduced at the beginning of section 3.
> * We thank the reviewer for pointing this out. We have revised this in our work.
>
>
> We appreciate the reviewer’s engagement with our work and we hope that we have addressed all the mentioned concerns. Please let us know if there are any more questions.

---

> > ### Comment · Reviewer_Azjn · 2023-11-21
> >
> > Thanks for detailed answering. Most of my questions are solved. But I still highly recommand the authors to include a separation result to make the paper more complete and the title less overclaimed. Based on this, I prefer to maintain my rating unchanged.

---

> > > ### Author Response · Authors · 2023-11-22
> > >
> > > We are pleased to hear that most of your questions have been addressed.
> > >
> > > Addressing the separation result is future work.
> > >
> > > Upon reviewing the FAQ from the Author Guide for ICLR 2024 (https://iclr.cc/Conferences/2024/AuthorGuide), modifications to the title and abstract during the rebuttal stage are not allowed. This policy is specified under the FAQ section titled "During Discussion Stages":
> > >
> > > “Following the same policy as ICLR 2023, we will not allow to change the title and abstract. [Updated on Nov. 15, 2023: For the camera-ready version, we will allow minor changes in the title and abstract so long as the revised version stays close to the original abstract submission; The revised version shouldn’t read like a different paper compared to your original abstract submission.]”
> > >
> > >
> > > However, should there be an exception or if title changes are allowed by ICLR, we would like to express our intention to modify the title along the lines of the options proposed below:
> > >
> > > "Expressivity of Spiking Neural Networks through the Spike Response Model"
> > >
> > > "Expressivity of Spiking Networks through the Spike Response Model in Temporal Coding"

---

> > > > ### Comment · Reviewer_Azjn · 2023-11-23
> > > >
> > > > Thanks for your response. The new titles are more suitable as they emphasize SRM rather than SNN alone. Both titles are acceptable.

---

### Official Review · Reviewer_3s5T · 2023-10-26

**Soundness:** 3 good
**Presentation:** 3 good
**Contribution:** 2 fair
**Rating:** 3
**Confidence:** 5

**Summary:**

This paper provides a theoretical investigation into the expressivity of SNNs. The theoretical framework relies on classical spike response models and neuron firing times, specifically focusing on SNNs employing linear-response-with-delay neuron models. The study begins by approximating the ReLU activation function using these spiking neurons. It then establishes that SNNs with such models possess at least the same approximation capacity as ANNs. Finally, the paper demonstrates that the maximum number of linear regions generated by a one-layer SNN grows exponentially with input dimension, outperforming ReLU-based ANNs.

**Strengths:**

1. This paper is technically sound, employing formulas appropriately. Additionally, the remarks provided assist readers in comprehending the paper.

2. The paper provides a comprehensive introduction and discussion of prior studies. Furthermore, it adequately addresses both its contributions and limitations.

**Weaknesses:**

1. About the novelty. The current version of the paper has not sufficiently clarified its novelty. While the study of SNNs for universal approximation of continuous functions is mentioned, the function classes and settings explored in this paper, while different, may still be considered as an incremental progress in this field.

2. About the quality. The title may be somewhat overstated. This paper demonstrates that SNNs have the capability to approximate ANNs, rather than fundamentally surpassing them. As a result, the potential advantages of constructing non-continuous functions and utilizing sparse computation are not fully explored. Essentially, this work places a strong emphasis on establishing a bound that demonstrates SNNs can approximate ANNs, rather than thoroughly investigating whether and to what extent SNNs outperform ANNs. Therefore, the overall significance of this work may be somewhat limited.

3. About the contributions. There are several assumptions or settings that may not be feasible in practical scenarios. It is crucial to address these limitations. Additionally, an important aspect to consider is the impact of timing length on the approximation complexity. This is essential as SNNs maintain a specific mapping from spike sequences to other sequences, which can be either continuous or non-continuous.

**Questions:**

1. Is there any unique difficulties for spiking neurons but not ReLU neurons?

---

> ### Author Response · Authors · 2023-11-16
>
> We thank the reviewer for the time and effort dedicated to the reviewing of our submitted manuscript. Below, we provide a detailed response organized under the same headings as those used in the review.
>
> ## Summary
> Firstly, we note that our work **doesn't involve the approximation of Sobolev and L2 function spaces.** Moreover, regarding the ReLU activation, it is important to highlight that our approach does not involve approximating the ReLU activation function. Rather, Theorem 2 in our paper shows that a two-layer SNN can express or emulate the ReLU activation.
>
> ## Weaknesses
> > About the novelty. The current version of the paper has not sufficiently clarified its novelty. While the study of SNNs for universal approximation of continuous functions is mentioned, the function classes and settings explored in this paper, while different, may still be considered as an incremental progress in this field.
> * We believe that the novelty is outlined in the Contributions paragraph within the Introduction Section. Firstly, we prove that the SNNs have the capability to generate continuous piecewise linear (CPWL) mapping. Moreover, Theorem 3 shows that SNNs can reproduce the output of any ReLU network. Our expressivity result in Theorem 3 implies that SNNs can essentially approximate any function with the same accuracy and complexity bounds as (deep) ANNs employing a piecewise linear activation function, given the response function satisfies some basic assumptions, which has not been demonstrated before.
>
> > About the quality. The title may be somewhat overstated. This paper demonstrates that SNNs have the capability to approximate ANNs, rather than fundamentally surpassing them. As a result, the potential advantages of constructing non-continuous functions and utilizing sparse computation are not fully explored. Essentially, this work places a strong emphasis on establishing a bound that demonstrates SNNs can approximate ANNs, rather than thoroughly investigating whether and to what extent SNNs outperform ANNs. Therefore, the overall significance of this work may be somewhat limited.
> * It is important to highlight that our approach doesn't approximate ANNs. Rather, Theorem 3 shows that SNNs can reproduce the output of any deep ReLU network. Theorem 4 and Example 1 shows the superior expressiveness of SNNs over ANNs. In fact, it highlights a characteristic of spiking neurons that distinguishes it significantly from a ReLU neuron, thereby illustrating differences in the structure of computations between SNNs and ANNs. But we acknowledge that this work doesn’t explore the benefits of SNNs for constructing non-continuous functions and it is planned as future work.
>
> > About the contributions. There are several assumptions or settings that may not be feasible in practical scenarios. It is crucial to address these limitations. Additionally, an important aspect to consider is the impact of timing length on the approximation complexity. This is essential as SNNs maintain a specific mapping from spike sequences to other sequences, which can be either continuous or non-continuous.
> * We acknowledge that our work utilizes a highly simplified version of the model, particularly by neglecting multi-spikes, sequential input, and refractoriness effects. While our setting might not be best suited for neuromorphic hardware, it is still applicable on digital hardware. This has been addressed in the limitations section. We recognize that exploring the effect of timing length on approximation complexity is an essential question for future research. The timing or simulation length can be a significant complexity measure. Further research in this direction is necessary to evaluate the complexity of SNNs using different measures, understanding their benefits, and drawbacks.
>
> ## Questions
> > Is there any unique difficulties for spiking neurons but not ReLU neurons?
> * The behaviour of spiking neurons is influenced by various parameters, and the analysis becomes intricate when considering the entangled effect of all these parameters on the output produced by the spiking network. Additionally, the results depend on the encoding scheme that one adopts. The introduced encoding scheme translates analog information into input firing times in a continuous manner. For tasks that involve modeling complex time patterns or a sequence of events with multiple spikes, sophisticated encoding schemes may be required, adding another level of difficulty to the analysis. However, our focus in this work lies on exploring the intrinsic capabilities of SNNs, rather than the specifics of the encoding scheme.
>
> We appreciate the reviewer’s engagement with our work and we hope that we have addressed all the mentioned concerns. Please let us know if there are any more questions.

---

> > ### Comment · Reviewer_3s5T · 2023-11-21
> > **Responses**
> >
> > Thanks for the detailed rebuttal.
> >
> > I notice that there are three points in the rebuttal.
> >
> > 1. "It is important to highlight that our approach doesn't approximate ANNs. Rather, Theorem 3 shows that SNNs can reproduce the output of any deep ReLU network." If so, this theorem is trivial since both SNN and ANN are universal approximators. In other words, SNN can approximate any function expressed by ANNs. The universal approximation properties of SNNs has been investigated in several studies, such as [1] [2].
> >
> > - [1] Networks of spiking neurons: The third generation of neural network models. 1996.
> > - [2] Theoretically provable spiking neural networks. NeurIPS' 2022.
> >
> > 2. The authors admitted that "while our setting might not be best suited for neuromorphic hardware, it is still applicable on digital hardware." However, this point is opposite to their claim in Impact on Page 2, in which "our findings prove that the low-power neuromorphic implementation of SNNs is an energy-efficient alternative to the computation performed by (ReLU-)ANNs without loss of expressive power."
> >
> > 3. This work employs a highly simplified version of the SNN model since "the analysis becomes intricate when considering the entangled effect of all these parameters on the output produced by the spiking network." In fact, there are several studies that attempt to explore the expressive advantages of non-trivial SNNs. There may have been a consensus that the expressive power of SNN originates from the intrinsic operations and parameters. Therefore, the contribution of this manner that simplifies SNNs to explore its expression advantages is relatively limited.
> >
> > - [3] Bifurcation spiking neural network. JMLR'2021.
> > - [4] On the algorithmic power of spiking neural networks. ITCS'2019.

---

> > > ### Author Response · Authors · 2023-11-22
> > >
> > > > "It is important to highlight that our approach doesn't approximate ANNs. Rather, Theorem 3 shows that SNNs can reproduce the output of any deep ReLU network." If so, this theorem is trivial since both SNN and ANN are universal approximators. In other words, SNN can approximate any function expressed by ANNs. The universal approximation properties of SNNs has been investigated in several studies, such as [1] [2]. [1] Networks of spiking neurons: The third generation of neural network models. 1996. [2] Theoretically provable spiking neural networks. NeurIPS' 2022.
> > >
> > > * The differences between our work and [1] have been highlighted in the beginning of the Related Work Section of the main paper. "In comparison to [1], we show that SNNs have the capability to realize the output of arbitrary ANNs with continuous piecewise linear (CPWL) activation functions and further specify the size of the network needed to achieve the associated realization, which has not been previously demonstrated. Moreover, we also study the expressivity of SNNs in terms of the number of linear regions and provide new insights on realizations generated by SNNs, that were not explored in [1]."
> > >
> > > * Moreover, the difference between [2] and our work has also been highlighted in the Related Work. “In [2], the authors investigate self-connection SNNs, demonstrating their capacity to efficiently approximate discrete dynamical systems. Our approach centers on precise spike timing, while theirs hinges on firing rates and includes a distinct model featuring self-connections, further setting their approach apart from ours.  Therefore, direct comparisons between our work and theirs might not be suitable.”
> > >
> > > > The authors admitted that "while our setting might not be best suited for neuromorphic hardware, it is still applicable on digital hardware." However, this point is opposite to their claim in Impact on Page 2, in which "our findings prove that the low-power neuromorphic implementation of SNNs is an energy-efficient alternative to the computation performed by (ReLU-)ANNs without loss of expressive power."
> > >
> > > * We have not implemented our approach on neuromoprphic hardware but we have implemented it on the digital hardware. This serves as a proof of concept, showcasing the applicability of our methodology as an alternative to the computation performed by ReLU-ANNs.

---

### Official Review · Reviewer_bNbs · 2023-10-28

**Soundness:** 3 good
**Presentation:** 3 good
**Contribution:** 3 good
**Rating:** 6
**Confidence:** 4

**Summary:**

This paper proves that SNNs generate continuous piecewise linear mappings when using the SRM spiking neuron model. It also shows that the maximum number of linear regions generated by a spiking neuron scales exponentially with respect to the input dimension, which is more powerful than the ReLU.

**Strengths:**

This paper is well-written and technically solid.

**Weaknesses:**

The proving of Theorem 1, and 2 are similar to those in [Yar17] and [SCC18]. It seems that the only difference is using different neurons.

**Questions:**

The universal approximation theorem of SNNs has been proved in [Maass, 1996c]. What is the difference between this paper and [Maass, 1996c]?

In fact, the SRM model is not the most popular spiking neuron model in deep SNNs. Why do the authors use the SRM model, rather than the LIF neuron in [Wu et al., 2018]?

---

> ### Author Response · Authors · 2023-11-16
>
> We thank the reviewer for the time and effort dedicated to the reviewing of our submitted manuscript. Below, we provide a detailed response organized under the same headings as those used in the review.
>
> ## Weaknesses
> > The proving of Theorem 1, and 2 are similar to those in [Yar17] and [SCC18]. It seems that the only difference is using different neurons.
> * The references [Yar17] and [SCC18] that the reviewer mentions have not been cited in the manuscript. Can you please provide more details on this?
>
> ## Questions
> > The universal approximation theorem of SNNs has been proved in [Maass, 1996c]. What is the difference between this paper and [Maass, 1996c]?
> * This reference [Maass, 1996c] has not been cited in the paper.  But assuming it corresponds to [1], the differences between our work and [1] have been highlighted in the Related Work Section of the main paper. In comparison to [1], we show that SNNs have the capability to realize the output of arbitrary ANNs with continuous piecewise linear (CPWL) activation functions and further specify the size of the network needed to achieve the associated realization, which has not been previously demonstrated. Moreover, we also study the expressivity of SNNs in terms of the number of linear regions and provide new insights on realizations generated by SNNs, that were not explored in [1].
>
> > In fact, the SRM model is not the most popular spiking neuron model in deep SNNs. Why do the authors use the SRM model, rather than the LIF neuron in [Wu et al., 2018]?
> * In line with our initial approach, we adopted the model choice based on [1] and we intend to expand our investigation to encompass multi-spike responses and refractoriness effects, thus, the choice of this model is appropriate and comprehensive. As mentioned in Section 2, page 4  "when the parameter $\delta$ is large, the simplified Spike Response Model, stemming from the linear response function and the constraint of single-spike dynamics, exhibits similarities to the integrate and fire model. Conversely, if $\delta$ is small, it resembles the leaky integrate and fire model.” As mentioned in the limitations section in the main paper, "our choice of model resides on theoretical considerations and not on practical considerations regarding implementation.” We acknowledge that there might be other models that are more apt for implementation purposes. Moreover, there are other works which in fact use the SRM model, for instance [2,3].
>
>
> [1] Wolfgang Maass. Noisy spiking neurons with temporal coding have more computational power than sigmoidal neurons. In Advances in Neural Information Processing Systems, volume 9. MIT Press, 1996.
>
> [2] I. M. Comsa, K. Potempa, L. Versari, T. Fischbacher, A. Gesmundo and J. Alakuijala, "Temporal Coding in Spiking Neural Networks with Alpha Synaptic Function," ICASSP 2020 - 2020 IEEE International Conference on Acoustics, Speech and Signal Processing (ICASSP), Barcelona, Spain, 2020, pp. 8529-8533, doi: 10.1109/ICASSP40776.2020.9053856.
>
> [3] B. Rueckauer and S. -C. Liu, "Conversion of analog to spiking neural networks using sparse temporal coding," 2018 IEEE International Symposium on Circuits and Systems (ISCAS), Florence, Italy, 2018, pp. 1-5, doi: 10.1109/ISCAS.2018.8351295.
>
> We appreciate the reviewer’s engagement with our work and we hope that we have addressed all the mentioned concerns. Please let us know if there are any more questions.

---

> ### Comment · Reviewer_bNbs · 2023-11-21
>
> I apologize for the mistakes I made when I did not notice the update and read the old version, which caused trouble for the authors, the reviewers, and the AC.
>
> > Please outline what has changed between the NeurIPS and ICLR versions
>
> I think there is not much difference between the key methods and contributions of the two versions. The concern of using the SRM model which can only fire at most one spike is still not solved. The neuron firing at most one spike is prone to causing the "dead neuron" problem that when no neurons spike, no learning occurs. As far as I know, most of the high-performance deep SNNs use the LIF or the Integrate-and-Fire (IF) neuron models and do not restrict the number of firing times. Thus, the contributions of this paper can be greatly improved if the widely-used LIF neuron, rather than the SRM model, is used. Meanwhile, I appreciate that the authors also recognize the importance of multi-spike neuron models as it "may yield an improvement on our results".
>
>
> > If you are not able to respond before the end of the author discussion period (in 2 days), I will need to ignore your review completely in my final evaluation
>
> My previous mistakes weaken the convincingness of my reviews. I apologize again to the authors, the reviewers, and the AC. Please feel free to ignore my review if needed.

---

> > ### Author Response · Authors · 2023-11-22
> >
> > Thank you for the apology and we appreciate the honesty of the reviewer.
> >
> > We want to clarify that while the linear Spike Response Model (SRM) remains a fundamental part of our methodology in both versions, there are significant differences in the results between the two versions. We have dedicated substantial efforts to refining and enhancing the technical aspects of our approach, resulting in a notable improvement in the overall outcomes. We have also included a lot of additional technical details in the current version to provide better clarity on our results.
> >
> > The dead neuron problem is addressed in the Remark 3 in Section 3.1.

---

> > > ### Comment · Reviewer_bNbs · 2023-11-23
> > >
> > > Thanks for your response. Although the work of the multi-spike neuron models, such as the LIF neuron, is not finished, the contributions of using the SRM model with firing at most one time can be the first step. I am willing to raise my score.
> > >
> > > At the same time, I also argee with the reviewer Azjn that the paper title should be changed (at the camera-ready version) to avoiding the over-claiming of contributions.

---

### Official Review · Reviewer_2Xv7 · 2023-11-02

**Soundness:** 3 good
**Presentation:** 2 fair
**Contribution:** 2 fair
**Rating:** 6
**Confidence:** 4

**Summary:**

The paper is a theoretical study of a model of spiking neurons, namely a linear Spike Response Model. It shows first that a network of such neurons is a continuous piecewise linear function, second that it can emulate any ReLU network, and third that the pieces of the piecewise function scales at most exponentially with the input dimension.

**Strengths:**

The paper is in a field that is attracting growing interest. Moreover, I find that the authors have chosen to address a quite significant question, broadly speaking, namely the question of whether spiking neural networks have any theoretical advantages over more conventional models. Furthermore, I find the analogies between the linear spike response model and the ReLU insightful, to some extent.

**Weaknesses:**

The paper studies only a specific spiking model, namely a linear SRM. However, the abstract and the main text suggest in multiple occasions that the paper's results apply to a much broader class of models, i.e. SNNs.
The title in particular suggests that the paper could rule out a comparative advantage in the expressive power of SNNs, but the paper has not studied the full class of SNN models, so it cannot possibly do that. In fact, other works have shown such advantages (see next paragraph), so the title's question has arguably already been answered in the literature.

The paper has not reviewed certain quite relevant prior results from the literature.
Other spiking neuron models, for example models with spike-based short-term plasticity, have been theoretically studied and shown to be more powerful than several non-spiking models in certain settings of temporal input. Moreover, that model showed in practice that networks of spiking neurons can actually outperform conventional models even in accuracy, not only in energy efficiency.
In addition, networks consisting partly of spiking neurons have been shown in both theory and practical benchmarks that they can learn to express not only _what_ has occurred in the input (classification), but also when (in the timing of the network's output), speeding up inference without trading off classification accuracy. This is another theoretical difference in expressivity from other neuronal models.
These recent but prior results have shown concrete advantages in the expressive power of spiking models in both theory and practice, but have they have not been mentioned in the present manuscript.

Similarly, other relevant works have shown that spiking neurons can represent multiple variables simultaneously, but they have also not been cited [3, 4].

These prior works take away some of the novelty of the present work. In addition, they suggest that today it is possible to couple theoretical insights on the expressive power of SNNs along with practical demonstrations of any differences - which is missing from the manuscript that is under review here. Only theoretical results are presented here.

Moreover, it appears difficult even for future researchers to translate these theoretical insights into practical outcomes, as the authors here have not provided any directions.

Furthermore, it appears to me that the results of the paper are not surprising. Using linear response functions with different gains at each input synapse, of course results in a piecewise linear combined function. Such a function of course can be shown equivalent to a certain combination of ReLUs. Having a different linear function at each synapse, of course can generate a larger number of linear pieces for the resulting function than a single ReLU. Or at least so it seems to me. Essentially, isn't the work merely based on expressing the input and the output as functions of time, but otherwise everything else is equivalent to conventional networks?

Lastly, that supposed difference between the spiking unit's and the ReLU perhaps is not significant. Isn't a spatially encoded input that is provided to a layer of multiple ReLUs equivalent to the temporal encoding and the SRM unit that the authors have assumed?


[1] Moraitis et al., "Optimality of short-term synaptic plasticity in modelling certain dynamic environments", arXiv 2021

[2] Jeffares et al., "Spike-inspired rank coding for fast and accurate recurrent neural networks", ICLR 2022

[3] Moraitis et al., "Spiking neural networks enable two-dimensional neurons and unsupervised multi-timescale learning", IJCNN 2018

[4] Izhikevich, "Polychronization: computation with spikes", Neural computation 2006

**Questions:**

Could the authors address the weaknesses in their responses here and in the manuscript? I am open to revising my evaluation if these issues are sufficiently addressed.

---

> ### Author Response · Authors · 2023-11-15
>
> We thank the reviewer for the time and effort dedicated to the reviewing of our submitted manuscript. Below, we provide a detailed response organized under the same headings as those used in the review.
>
> ## Weaknesses
> > The paper studies only a specific spiking model, namely a linear SRM. However, the abstract and the main text suggest in multiple occasions that the paper's results apply to a much broader class of models, i.e. SNNs.
>
> * We acknowledge the reviewer that the results in the paper hold in the case of linear SRM where each neuron spikes at most once. However, we have mentioned in the Abstract, Contributions, Section 2 and Discussion Section that the results hold only in the case of linear SRM. Can you indicate the specific locations in the manuscript where the text implies the above statement? We will adapt the text accordingly based on your feedback.
>
> > The paper has not reviewed certain quite relevant prior results from the literature. Other spiking neuron models, for example models with spike-based short-term plasticity, have been theoretically studied and shown to be.....
>
> * We appreciate the valuable insights provided by the reviewer and have taken their comments into consideration. The related works mentioned, particularly those involving spiking neuron models with spike-based short-term plasticity and networks of spiking neurons that outperform conventional models in both accuracy and energy-efficiency are indeed noteworthy. In response to the reviewer’s feedback, we have incorporated these relevant works into the manuscript (highlighted in red). These results provide intriguing insights on the expressivity of spiking networks, showcasing their ability to excel in certain settings of sequential temporal input, and predict environments with continuous random dynamics. However, we would like to point out that the direct comparison between our work and these related studies is not suitable due to differences in the chosen models, objectives and methodologies. In the simplest of settings, there remains a lack of comprehensive theory that completely quantifies the approximation capabilities of spiking neural networks. We consider addressing these questions essential and it forms the central focus of this work. While universal approximation results exist for spiking networks, our work introduces the idea of linear regions specific to spiking networks and provides the necessary complexity bounds for realizing different functions. These notions of expressivity hold particular significance from a mathematical standpoint, providing insights into the mathematical capabilities of spiking networks in realizing continuous piecewise linear (CPWL) functions. These notions shed light on the mathematical capacity of a spiking network whereas practical performance is contingent on the efficiency of the learning algorithm.
>
> > The title in particular suggests that the paper could rule out a comparative advantage in the expressive power of SNNs.....
>
> * We acknowledge your concern regarding the title. Unfortunately, to the best of our knowledge, the title change is not allowed by the ICLR anymore.
>
> > These prior works take away some of the novelty of the present work. In addition, they suggest that today it is possible to couple theoretical insights on the expressive power of SNNs along with practical demonstrations of any differences - which is missing from the manuscript that is under review here. Only theoretical results are presented here. Moreover, it appears difficult even for future researchers to translate these theoretical insights into practical outcomes, as the authors here have not provided any directions.
>
> * We would like to emphasize that this research is primarily focused on theoretical aspects rather than practical applications. Our primary goal in this investigation is to explore the capabilities of SNNs in the simplest of settings, recognizing that comprehending the simple scenarios not only aids but is also essential for comprehending intricate settings. We have conveyed this in the limitations section. To translate the theoretical result into practical outcomes, one can adapt the training algorithm implemented in [1] to our linear SRM model. We are actively working on the practical implementation and it is currently in progress.
>
> [1] Göltz, J., Kriener, L., Baumbach, A. et al. Fast and energy-efficient neuromorphic deep learning with first-spike times. Nat Mach Intell 3, 823–835 (2021).
>
>
> **The other comments are continued below.**

---

> ### Author Response · Authors · 2023-11-15
>
> **Previous comment continued**
>
> > Furthermore, it appears to me that the results of the paper are not surprising. Using linear response functions with different gains at each input synapse, of course results in a piecewise linear combined function. Such a function of course can be shown equivalent to a certain combination of ReLUs. Having a different linear function at each synapse, of course can generate a larger number of linear pieces for the resulting function than a single ReLU. Or at least so it seems to me.
>
> * Our paper rigorously establishes that SNNs (linear SRM) generate continuous piecewise linear mapping and that SNNs can reproduce the output of any ReLU network. While it might seem intuitive that using different gains at each input synapse could result in a piecewise linear function, the mathematical details and complexity bounds provided in our work have not been demonstrated in any prior work. The intuition expressed by the reviewer aligns with the general understanding that continuous piecewise linear functions can be shown to be equivalent to a certain combination of ReLU’s units. However, our contribution lies in the detailed examination of the structure and complexity of SNNs in realizing these functions.  For instance, in the proof of Theorem 2 in the Appendix A.3, it is demonstrated that a one layer SNN cannot realize the ReLU activation, providing unique insights into the structure of CPWL functions that may not be efficiently realized by SNNs compared to ReLU ANNs under the given assumptions on the response function. Regarding Theorem 4, it not only provides a tight bound on the number of linear regions into which the input domain is divided by an SNN unit but also describes the necessary conditions required to achieve it. Comprehensive details can be found in Section A.2 of our paper, including an explicit computation of the boundaries of linear regions through Example A.2.
>
> > Essentially, isn't the work merely based on expressing the input and the output as functions of time, but otherwise everything else is equivalent to conventional networks?
>
> * Since we consider only the feedforward networks, this allows us to express layerwise computations performed in a ReLU ANN but using an SNN. However, as mentioned in Remark 1, the internal dynamics governing our SNN model significantly deviate from those of traditional ANNs. In essence, while the structural framework may bear resemblance to conventional ANNs in terms of input-output mapping, the unique characteristics of SNNs as outlined in our analysis leads to the differences in the structure of computations between SNNs and ANNs.
>
> >  Lastly, that supposed difference between the spiking unit's and the ReLU perhaps is not significant. Isn't a spatially encoded input that is provided to a layer of multiple ReLUs equivalent to the temporal encoding and the SRM unit that the authors have assumed?
>
> * If we understand your question correctly, then it’s important to note that our work reveals  specific scenarios where differences in the expressive power exist. Example 1 illustrates that a function $f$ can be efficiently realized by a single spiking unit, whereas a two layer ANN would be required to express the same function. Essentially, Theorem 4 illustrates this point that a certain function can be realized by a single SNN unit whereas a deep ReLU (with multiple layers) ANN would be required to approximate the same function, highlighting a distinct advantage of SNNs in certain contexts. This contradicts your suggested equivalence between multiple ReLU’s in a layer and a spiking unit.
>
> We believe that our results contribute significantly to the understanding of expressivity of SNNs and their comparison to conventional ANNs. It also provides a foundation for future explorations into more complex settings, for instance, incorporating multi-spikes and coupled effect of response and threshold functions on the approximation results. We hope that the reviewer recognizes and takes these implications into account during the evaluation process.
>
> We appreciate the reviewer’s engagement with our work and we hope that we have addressed all the mentioned concerns. Please let us know if there are any more questions.

---

> ### Comment · Reviewer_2Xv7 · 2023-11-21
>
> I would like to thank the authors for their responses. These are helpful. I am convinced that the paper brings useful insights to the SNN field.
>
> However, there is one very important broader issue that remains unresolved, with two sub-issues. The paper as it is currently written asserts that its results are broader than they actually are.
>
> **The first subissue**, again, is that the title is not acceptable. The paper cannot actually pose the title's question about SNNs, because that question has already been answered in the literature, as the authors show with the correctly added references to their revised version. Moreover, the paper does not attempt to disprove any hypothesis about the broad category of networks that are SNNs, but rather only focuses on SRMs, contrary to what the title implies.
>
> > Unfortunately, to the best of our knowledge, the title change is not allowed by the ICLR anymore.
>
> The FAQ from the Author Guide https://iclr.cc/Conferences/2023/AuthorGuide reads:
>
> *Q. Can we change the title of a paper during the rebuttal?*
>
> *Yes, you can change the title, abstract, and the paper’s content, including supplementary materials. But make sure any modifications are clearly communicated to the reviewers and the area chair, so that they can efficiently review the modified version of your paper. The set of authors cannot be changed, but the order can be changed.*
>
> **The second sub-issue** is that similarly, throughout the paper there are statements about SNNs, which are a super-set of SRMs and therefore they are not supported by any of the results, which only apply to SRMs specifically.
> The problematic sentences are too many to quote them all, but they are almost all the sentences that mention "SNNs". One example from the introduction: *Specifically, we aim to determine if SNNs possess the same level of expressiveness as ANNs*
>
> If these issues could be resolved, I would be open to raising my score.

---

> ### Author Response · Authors · 2023-11-22
>
> Thank you for your feedback and we are pleased to hear that most of your questions have been addressed.
>
> > The first subissue, again, is that the title is not acceptable. The FAQ from the Author Guide https://iclr.cc/Conferences/2023/AuthorGuide reads:
> Q. Can we change the title of a paper during the rebuttal?
> Yes, you can change the title, abstract, and the paper’s content, including supplementary materials. But make sure any modifications are clearly communicated to the reviewers and the area chair, so that they can efficiently review the modified version of your paper. The set of authors cannot be changed, but the order can be changed.
>
> * The FAQ link shared by the reviewer refers to **ICLR 2023**. Upon reviewing the FAQ from the Author Guide for ICLR 2024 (https://iclr.cc/Conferences/2024/AuthorGuide), modifications to the title and abstract during the rebuttal stage are not allowed. This policy is specified under the FAQ section titled "During Discussion Stages":
> “Following the same policy as ICLR 2023, we will not allow to change the title and abstract. [Updated on Nov. 15, 2023: For the camera-ready version, we will allow minor changes in the title and abstract so long as the revised version stays close to the original abstract submission; The revised version shouldn’t read like a different paper compared to your original abstract submission.]”
>
> However, should there be an exception or if title changes are allowed by ICLR, we would like to express our intention to modify the title along the lines of the options proposed below:
>
> "Expressivity of Spiking Neural Networks through the Spike Response Model"
>
> "Expressivity of Spiking Networks through the Spike Response Model in Temporal Coding"
>
> > The second sub-issue is that similarly, throughout the paper there are statements about SNNs, which are a super-set of SRMs and therefore they are not supported by any of the results, which only apply to SRMs specifically. The problematic sentences are too many to quote them all, but they are almost all the sentences that mention "SNNs". One example from the introduction: Specifically, we aim to determine if SNNs possess the same level of expressiveness as ANNs
>
>
>  * To address your concern about “SNNs”, we have revised our manuscript. In particular, beginning from the Contributions paragraph, we have replaced the abbreviation “SNNs” with “LSRMs” (in red) throughout the paper where appropriate. LSRMs now stand for Linear Spike Response Models, making it explicit that our results specifically pertain to this subset of SNNs. By incorporating this change, we aim to clarify that our findings are applicable only to Linear Spike Response Models, providing a more accurate representation of the scope of our results.
>
> We hope that the applied revisions are to the satisfaction of the reviewer. If there is anything else, please let us know.

---

> > ### Comment · Reviewer_2Xv7 · 2023-11-22
> >
> > Dear Area Chair,
> >
> > Can the authors' suggested title revision be approved?
> >
> > If so, then I would recommend acceptance of the paper, as it provides a useful analogy to a spiking model, supported by theory. The authors have addressed my most important concerns.
> >
> > However, my recommendation would not be strong, mainly because of the lack of any empirical contributions or even recommendations for future practical impact of the results, because of the somewhat unsurprising results given the assumptions, and because the paper chose a spiking model that is minimally different from non-spiking ones, so the results were destined to be somewhat underwhelming.
> > On the other hand, this makes it useful by being simple to understand, despite its theoretical rigor.
> >
> > Based on these, I raise my score, with the recommendation being conditional on a title change.
> >
> > Dear authors, thank you for the improvements.

---

### Meta-Review · Area_Chair_z9iV · 2023-12-09

**Metareview:**

This paper considers the expressivity of spiking neural networks under the Linear Spike Response Model (LSRM), showing that such networks generate continuous piecewise linear mappings and analyzing the number of linear regions generated. The reviewers agree that this is an interesting topic, but have reservations about the relevance of the LSRM model to yield conclusions applicable to spiking neural networks in general, the way in which the results are presented in the present paper (the authors have agreed to change the title), and the extent to which the results establish relatively intuitive statements. Given this concerns, I must recommend rejection.

I will stress the following elements in particular:
- While the authors have agreed to change the title, the paper overall is still overselling the results considerably. The first sentence of the abstract claims a connection that is never remotely supported in the paper: "This article studies the expressive power of spiking neural networks with firing time-based information encoding, highlighting their potential for future energy efficient AI applications when deployed on neuromorphic hardware." The last sentence of the abstract does similarly: "Our results further extend the understanding of the approximation properties of spiking neural networks and open up new avenues where spiking neural networks can be deployed instead of artificial neural networks without any performance loss." This is a theory paper, which is fine, but if the authors are claiming anything beyond theoretical relevance, then in my opinion they need to work with (i) a more real-world set of assumptions, or (ii) at least some empirical results that show how much their theoretical results extend to a broader set of conditions.
- The result of exponentially many linear regions also seems to me to be oversold considerably. While the authors claim this shows that spiking neurons can express in a shallow network the same functions as in a deep ReLU network, it's actually just showing that a 1-layer LSRM network is similar to a 2-layer ReLU network, which is perhaps a semantic distinction in how "depth" is defined. (Many authors treat a single hidden layer as "depth-1" rather than "depth-2".)

**Justification For Why Not Higher Score:**

See "Additional Comments On Reviewer Discussion".

**Justification For Why Not Lower Score:**

n/a

---

### Decision · Program_Chairs · 2024-01-16

Reject